# Single-shot self-supervised object detection in microscopy

Benjamin Midtvedt[1], Jesús Pineda [1], Fredrik Skärberg[1], Erik Olsén [2], Harshith Bachimanchi [1], Emelie Wesén[3], Elin K. Esbjörner [3], Erik Selander [4], Fredrik Höök [2], Daniel Midtvedt[1] & Giovanni Volpe [1]✉

Object detection is a fundamental task in digital microscopy, where machine learning has made great strides in overcoming the limitations of classical approaches. The training of state-of-the-art machine-learning methods almost universally relies on vast amounts of labeled experimental data or the ability to numerically simulate realistic datasets. However, experimental data are often challenging to label and cannot be easily reproduced numerically. Here, we propose a deep-learning method, named LodeSTAR (Localization and detection from Symmetries, Translations And Rotations), that learns to detect microscopic objects with sub-pixel accuracy from a single unlabeled experimental image by exploiting the inherent roto-translational symmetries of this task. We demonstrate that LodeSTAR outperforms traditional methods in terms of accuracy, also when analyzing challenging experimental data containing densely packed cells or noisy backgrounds. Furthermore, by exploiting additional symmetries we show that LodeSTAR can measure other properties, e.g., vertical position and polarizability in holographic microscopy.

The study of biological systems often requires detecting individual objects, from microorganisms to biomolecules[1,2]. For example, individual object tracking enables the study of the life cycle and proliferation of single microorganisms[3] as well as the mobility of inter- and intracellular particles[4,5]. On the molecular level, high-contrast imaging of biological processes with unprecedented spatio-temporal resolution has been made possible by single-molecule fluorescence and super-resolution microscopy[6]. However, like many tasks in computer vision, object detection is surprisingly difficult and it is now often the limiting factor in the analysis of microscopic images.

Recently, deep learning has been successfully employed to improve object detection in microscopy, outperforming more standard methods, e.g., to track particles in noisy images[7,8], to push the limits of super-resolution fluorescence[9], and to quantitatively characterize sub-wavelength particles[10]. The most widely used deep-learning methods use supervised learning, where a neural network is trained to solve a particular problem using large amounts of high-quality training data consisting of input data and corresponding expected results (ground truth). Obtaining these datasets represents the effective bottleneck in the application of deep learning to microscopic object detection[2]. While public datasets are available, they are often inadequate to represent the idiosyncrasies of any specific experimental sample and setup. Thus, acquiring the needed experimental datasets in house is often the only viable option, but it comes with its own burdens in terms of time and effort. As a consequence, most deep-learning methods for object detection rely on synthetic data[7–11]. However, accurate synthetic replication of experimental data is very challenging, even for relatively simple transmission microscopes[2].

Even if a sufficiently large dataset can be collected, determining the corresponding ground truth with sufficient accuracy can be even more challenging. Human-derived annotation of experimental data is highly labor-intensive and prone to inconsistencies[12], especially when dealing with high-noise data or when sub-pixel precision is required.

[1]Department of Physics, University of Gothenburg, Gothenburg, Sweden. [2]Department of Physics, Chalmers University of Technology, Gothenburg, Sweden. [3]Department of Biology and Biological Engineering, Chalmers University of Technology, Gothenburg, Sweden. [4]Department of Marine Sciences, University of Gothenburg, Gothenburg, Sweden. ✉e-mail: giovanni.volpe@physics.gu.se

Automatically-generated annotations can be employed by mimicking an existing method, but this relies on a prior ability to algorithmically analyze the data—and the resuting deep-learning method is unlikely to outperform the original method. Finally, numerically-generated data naturally come with the exact ground truth used to generate them, but their applicability is limited to experimental setups that can be recreated numerically, which is only possible in some cases.

Here, we tackle these issues by developing a novel deep-learning approach, named LodeSTAR (Localization and detection from Symmetries, Translations, And Rotations), that exploits the inherent symmetries of object detection to enable training on extremely small datasets (down to a single image of the object) without ground truth. We demonstrate that a single training image is sufficient to train LodeSTAR to outperform standard methods in terms of accuracy, while simultaneously providing robust detection in complex experimental conditions such as densely-packed or noisy images. Furthermore, beyond in-plane object detection, we demonstrate that it is possible to exploit additional symmetries to measure other object properties for which no widespread standard methods are available, e.g., the vertical position in interference holography exploiting the propagation symmetry of the image in Fourier space and the object polarizability by exploiting the scaling symmetry of the signal strength of the image.

## Results

### LodeSTAR overview

LodeSTAR builds on geometric deep learning[13] and the recent surge of self-supervised object tracking methods[14–21] to create a self-supervised (or more precisely, self-distillative) object-detection neural network optimized for microscopy data. Specifically, we exploit the fact that a neural network that is equivariant to rotations and translations (i.e., a neural network for which a roto-translational transformation of the input image produces an equivalent roto-translation of the prediction) operates as an object detector (see Methods, "Theory of geometric self-distillation"). A limitation for general objects is that the exact part of the object that is detected cannot be controlled; however, if the object has a well-defined center (by having at least two axes of symmetry), such a neural network will find the exact center of the object. Building on this insight, we design a novel neural-network architecture that uses global weighted pooling to become inherently translation equivariant (see Methods, "Neural network architecture"). We also design a novel unsupervised training procedure that trains the neural network to become fully roto-translation equivariant. This procedure feeds the neural networks with transformed views of the same image of a single object, and trains it to predict positions that are equivariant with the transformations (see Methods, "Neural network training").

### Geometric self-distillation

LodeSTAR is designed to exploit the symmetries inherent in the detection of microscopic objects. For example, even if we do not know the absolute position of an object, we can say for sure that, if we translate the object image by a certain amount, its position gets also translated by the same amount—and similarly for rotations and reflections. This is known as an equivariance. In fact, for a object whose image has a well-defined center of symmetry, such as that shown in Fig. 1a, a prediction of the position of the object that is equivariant to the Euclidean group (translations, rotations, and reflections) can be shown to necessarily locate the center of the object (see Methods, "Theory of Geometric Self-Distillation").

LodeSTAR trains a neural network to achieve equivariance, i.e., an exact correspondence between the transformation applied to its input image (e.g., a translation by a certain amount) and the effect this has on the output prediction (e.g., a translation of the predicted object position by the same amount). We start with an image of a single object (Fig. 1a). By applying a transformation from the Euclidean group to this

image, we create several transformed images (e.g., the two images in Fig. 1b). A neural network (Fig. 1c) predicts the position of the object within each image (Fig. 1d, which will be more thoroughly explained in the next paragraph), yielding two predictions of the $x$ and $y$ coordinates of the object, one for each transformation (Fig. 1e). To compare these predictions, we back-transform the predicted coordinates to determine their corresponding positions in the original image (Fig. 1f). Then, if the neural network is equivariant to the applied transformations, the two predictions should perfectly overlap, while any deviation between these two prediction can be used to train the neural network (i.e., to adjust the neural network by updating its internal weights to minimize the distance between the two predictions).

We now turn our attention to the details of the neural network employed by LodeSTAR. While most architectures are compatible with LodeSTAR, it is convenient to use a completely translation-equivariant neural network (e.g., a fully convolutional neural network). This presents a two-fold advantage. First, having inherent translation-equivariance, the neural network does not need to learn the equivariance during training, which significantly reduces the required complexity of the model (while rotation-equivariance and reflection-equivariance still need to be learned). Second, as will become clear in the following sections, this will permit us to use the network to detect multiple objects while still training on a single image of the object.

As schematically shown in Fig. 1c (details are described in the Methods, "Neural network architecture"), we use a fully convolutional neural network, with two convolutional layers, one max-pooling layer, and seven additional convolutional layers, the last of which outputs three channels, $\Delta x$, $\Delta y$, and $\rho$. The outputs of this neural network are further analyzed, as shown Fig. 1d (this analysis also needs to be translation-equivariant). The vectors $(\Delta x, \Delta y)$ estimate the direction and distance from each pixel to the object (blue arrows in upper left panel of Fig. 1d; far away from the object, the lengths of these vectors go to zero because LodeSTAR only attempts to predict the position where the weight map $\rho$ is non-zero.). By adding to each of these vectors the respective pixel position, we retrieve a map of predictions of the object position relative to the center of the image. During evaluation the network transforms the predictions to be relative to the top left corner for convenience (blue markers in the upper right panel of Fig. 1d). The $\rho$ channel provides a weight map (normalized to sum to one) corresponding to the probability of finding the center of the object near each pixel (bottom panel in Fig. 1d). The final prediction of the object position is obtained by an average of the estimated object positions weighted by the weight map.

LodeSTAR manages to train this network using a single image of the object. Moreover, thanks to the small size of the neural network, it can be trained fully from scratch in an order of $10^4$ mini-batches, which takes a few minutes, even without a graphics processing unit. Finally, LodeSTAR can be trained with small batch-sizes, and as such needs less than a gigabyte of runtime memory (see details in the Methods, "Neural network training").

### Detecting a single object

We start by considering the performance of LodeSTAR on the simplest case: a point object (e.g., the image of a single molecule obtained from a fluorescence microscope). We simulate $10^4$ images of point objects with signal-to-noise ratio (SNR) between 2 and 20 using the Python library DeepTrack 2.1[2]. We use a single one of these images (inset in Fig. 1g, SNR = 10) to train LodeSTAR, using 5000 mini-batches of 8 samples. LodeSTAR achieves a sub-pixel root mean square error (RMSE) for all SNRs and a RMSE < 0.1 px for SNR > 5 (blue circles in Fig. 1g). Strikingly, LodeSTAR performs well also far from the SNR at which it is trained.

In fact, we find that LodeSTAR achieves a near optimal performance. We evaluate the optimal performance by calculating the Cramer-Rao (CR) lower bound on the localization error[22] (see Methods,

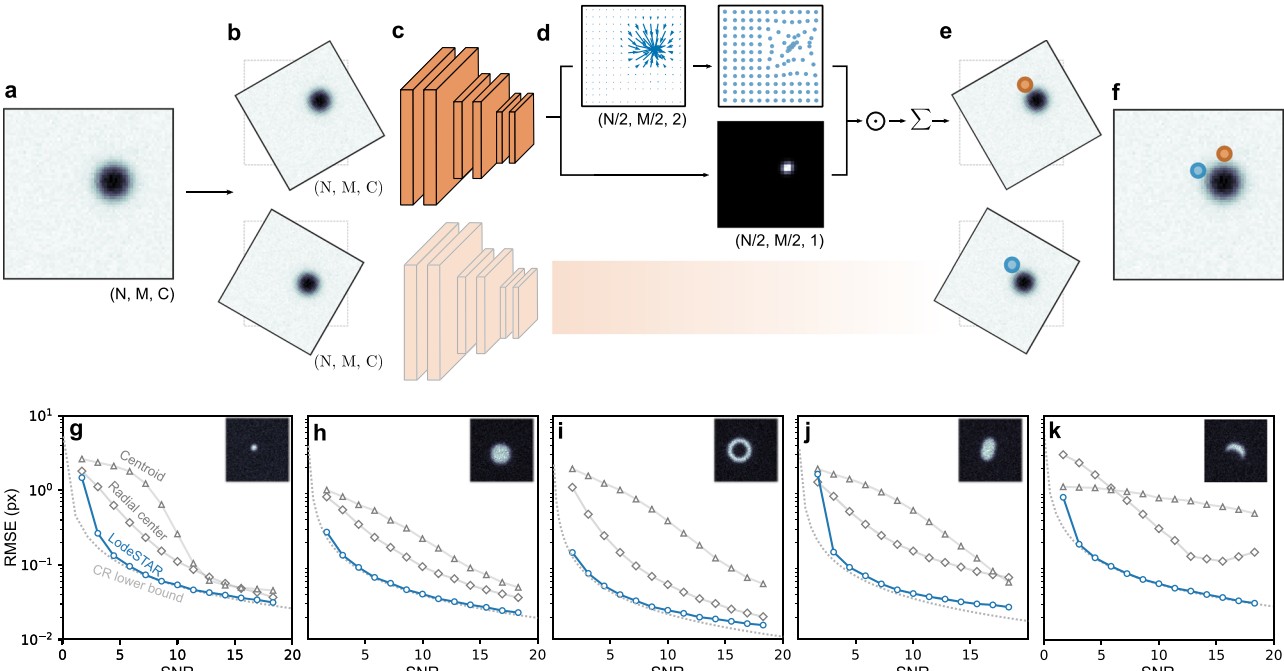

**Fig. 1 | LodeSTAR single-shot training and performance. a** Example image of a single particle used to train the neural network ($N \times M$ pixels, $C$ color channels). **b** Two copies of the original image transformed by translations and rotations. **c** The transformed images are fed to a convolutional neural network. **d** The neural network outputs two tensors (feature maps), each with $N/2 \times M/2$ pixels: One (top) is a vector field where each pixel represents the direction and distance from the pixel itself to the object (top left, blue arrows), which is then transformed so that each pixel represents the direction and distance of the object from the center of the image (top right, blue markers). The other tensor (bottom) is a weight map (normalized to sum to one) corresponding to the contribution of each element in the top feature map to the final prediction. **e** These two tensors are multiplied together and summed to obtain a single prediction of the position of the object for each

transformed image. **f** The predicted positions are then converted back to the original image by applying the inverse translations and rotations. The neural network is trained to minimize the distance between these predictions. **g–k** LodeSTAR performance on 64 px × 64 px images containing different simulated object shapes: **g** point particle, (**h**) sphere, (**i**) annulus, (**j**) ellipse, and (**k**) crescent. Even though LodeSTAR is trained on a single image for each case (found in the corresponding inset), its root mean square error (RMSE, blue circles) approaches the Cramer-Rao (CR) lower bound (dotted gray line), and significantly outperforms traditional methods based on the centroid[23] (gray triangles) or radial symmetry[24] (gray diamonds), especially at low signal-to-noise ratios (SNRs). Interestingly, even in the crescent case (**k**), where there is no well-defined object center, LodeSTAR is able locate it to within a tenth of pixel.

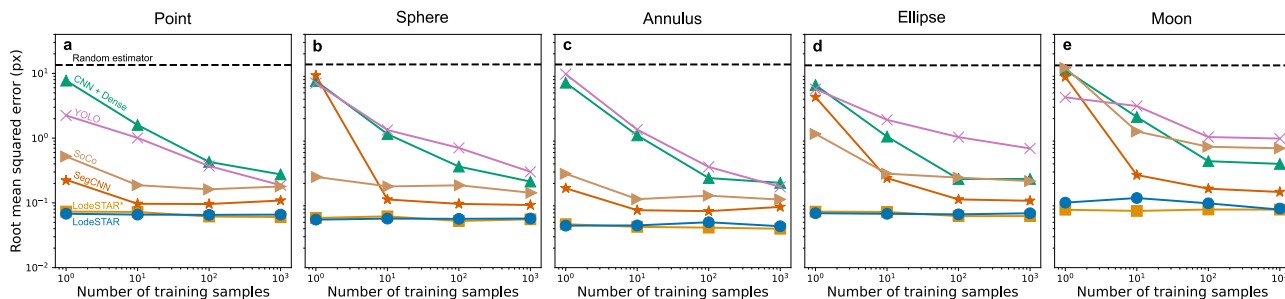

**Fig. 2 | Evaluation positioning accuracy of object detection methods.** The root mean squared error of the position accuracy for six methods, a CNN with a dense top [42], YOLOv4[25], SoCo[18], a segmentation CNN[7], LodeSTAR* (which is the architecture of LodeSTAR trained in a supervised manner), and LodeSTAR. Each model was trained according to the recommendations of the corresponding paper. The performance of the model was evaluated on a separate validation set of 1000

images during training, to ensure that the model did not under- or overtrain. We evaluate these over a range of sizes of training sets, from 1 datapoint to 1000 datapoints, on five shapes: **a** a point particle, (**b**) a spherical particle, (**c**) an annulus, (**d**) an ellipse, and (**e**) a crescent moon shape. LodeSTAR outperforms all other methods at all sizes of training sets. In fact, LodeSTAR reaches optimal performance using just one datapoint for training.

"Evaluating the Cramer-Rao lower bound" for more details). The CR lower bound defines the optimal performance any estimator can achieve based on the information content in the image. LodeSTAR manages to approach the CR lower bound for most SNRs, only falling short for very low SNRs (Fig. 1g).

In contrast, two of the most standard object localization methods, i.e., the centroid method[23] (gray triangles in Fig. 1g) and the

radial-center method[24] (gray diamonds in Fig. 1g) are unable to reach the CR lower bound in any experiment. While the radial-center method achieves sub-pixel accuracy over the whole range of SNRs, it is consistently outperformed by LodeSTAR. The radial-center method approaches LodeSTAR's performance for high SNRs, but is not competitive at lower SNRs. Overall, these results are consistent with the literature, where deep-learning-based methods have been

shown to outperform traditional methods especially at mid-to-low SNRs[2,7,8].

In contrast to traditional methods that are optimized for certain sets of object shapes, LodeSTAR can detect objects of arbitrary shapes. We show some examples in Fig. 1h–k, where we present the results for a sphere (Fig. 1h), an annulus (Fig. 1i), an ellipse (Fig. 1j), and a crescent (Fig. 1k). Overall, LodeSTAR achieves a sub 0.1 px error for the majority of the SNR range in all cases. LodeSTAR also nearly reaches the CRLB in all cases. While we can expect LodeSTAR to perform well on symmetric objects, such as the sphere (Fig. 1h) and the annulus (Fig. 1i), it is an important confirmation of the generality of LodeSTAR that it also works for non-symmetric objects, such as the ellipse (Fig. 1j), which has only two axes of symmetry instead of full radial symmetry, and the crescent (Fig. 1k), which has only one axis of symmetry. Supplementary Movie 1 demonstrates LodeSTAR locating the objects at various orientations and noise levels.

**Table 1 | LodeSTAR has discriminative power**

| Model particle | Point | Sphere | Annulus | Ellipse | Crescent moon |
|---|---|---|---|---|---|
| Point | **0.02** | 31.44 | 70.47 | 12.82 | 1590.37 |
| Sphere | 857.96 | **0.05** | 58.66 | 1.13 | 1025.23 |
| Annulus | 58.60 | 26.94 | **0.07** | 27.45 | 750.94 |
| Ellipse | 758.71 | 0.85 | 75.16 | **0.12** | 21.77 |
| Crescent | 13.97 | 11.47 | 18.32 | 10.10 | **0.18** |

Each model trained by LodeSTAR is significantly more self-consistent (lower average weighted variance) when presented data from its training distribution (bold font), compared to when it is evaluated on different data. This demonstrates that LodeSTAR has acquired discriminative power.

We also compare LodeSTAR to alternative deep-learning methods for object detection and localization, as a function of the number of training images. Namely, YOLOv4[25], SoCo[18], DeepTrack 1[8], a segmentation CNN[7], and the LodeSTAR architecture trained supervised. We find that LodeSTAR, trained on just one image, outperforms all other methods regardless of the size of the training set. The details are shown in Fig. 2.

We turn now our attention to the discriminative power of LodeSTAR, i.e., its ability to learn about the specific shape used in its training. Such discriminative power is important for heterogeneous samples or just to minimize false positives. In order to do this, we consider the distribution of predicted positions of the object in view preceding the global pooling operation, i.e., the feature map (Fig. 1d). Specifically, we measure the self-consistency of the model by calculating the weighted variance of the feature map. We expect that the neural network is highly self-consistent when evaluated on images similar to the training data, while less so when evaluated on something distinctly different. As can be seen in Table 1, the variance is several orders of magnitude lower when the model analyzes an image drawn from the distribution on which it has been trained. This demonstrates that LodeSTAR acts differently when presented data similar to the training distribution compared to data from another distribution, which clearly shows that LodeSTAR has acquired discriminative power.

## Detecting multiple objects

Thanks to the translation-equivariant design of the neural network, LodeSTAR trained on a single object image as described in the previous section can immediately be used to detect multiple objects without any additional training. In fact, since the receptive field of the convolutional network is limited, additional objects in view (such as in Fig. 3a) are analyzed largely independently by the neural network.

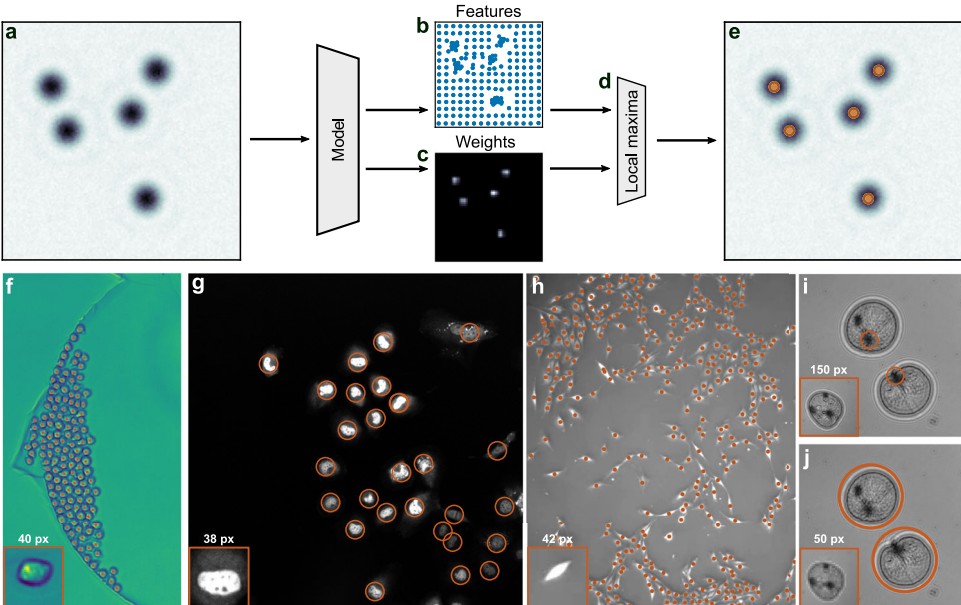

**Fig. 3 | LodeSTAR analysis of images with multiple objects. a** Example image with multiple objects to be detected by LodeSTAR (LodeSTAR is still trained on a single object image, as described in Fig. 1). **b** LodeSTAR returns clustered predictions of object positions and (**c**) a weight map representing the likelihood of finding a object near each pixel. **d** An estimation of the local density of object detections is multiplied by the weight map to obtain a detection map, whose local maxima are considered object detections (orange markers in **e**). **f–i** Examples of applications of LodeSTAR to experimental data that present different challenges. In all cases, LodeSTAR is trained on the single crop shown in the respective inset and then applied to the whole time-series. See also the corresponding Supplementary Movies 2–6. **f** LodeSTAR finds the positions of mouse hematopoietic stem cells (red markers), achieving an F1 score of 0.98 rate despite the dense sample (data from[12]). **g** LodeSTAR identifies human hepatocarcinoma-derived cells (red circles), achieving an F1 score of 0.97, despite the high variability between cells (data from[12]). **h** LodeSTAR detects pancreatic stem cells (red markers), an F1 score of 0.95, despite the densely packed sample and the high variability between cells (data from[12]). **i** LodeSTAR detects the plankton *Noctiluca scintillans*. In this case, LodeSTAR detects the optically dense area of the tentacle attachment point (red circles). **j** Interestingly, if the data is downsampled by a factor of 3 (so that the training image is 50 px × 50 px instead of 150 px × 150 px) before training and evaluation, the model finds the cell as a whole.

**Table 2 | LodeSTAR outperforms other selft-supervised and low-shot methods**

|  | Dataset 1 | Dataset 2 | Dataset 3 |
|---|---|---|---|
| SoCo | 0.85 | 0.84 | 0.48 |
| FSDet | 0.73 | 0.75 | 0.59 |
| InstanceLoc | 0.82 | 0.67 | 0.43 |
| DETReg | 0.71 | 0.61 | 0.44 |
| LodeSTAR | **0.98** | **0.97** | **0.95** |

F1-score of self-supervised and low-shot object detection methods evaluated on the datasets in the cell-tracking challenge BF-C2DL-HSC (dataset 1, Fig. 3f), Fluo-C2DL-Huh7 (dataset 2, Fig. 3g), and PhC-C2DL-PSC (dataset 3, Fig. 3h)[12]. The best scores are highlight in bold. LodeSTAR significantly outperforms the other methods in each experiment.

Taking advantage of these observations, we can bestow LodeSTAR with the capability to detect multiple objects, simply by removing the weighted global pooling layer (i.e., the final multiplication in Fig. 1e) and operating directly on the feature-maps themselves, i.e., on the predicted object position map (Fig. 3b) and on the weight map (Fig. 3c). By multiplying a measure of the local density of object positions by the weight map (Fig. 3d), we obtain a detection map whose local maxima represent the positions of the detected objects. After detection, the exact location of the object is determined by a weighted average of the local region around the detection, analogous to the single-object case, finally yielding the position of each object (Fig. 3e). See also details in Methods, "Particle detection criteria".

### Validation with experimental data

We now apply LodeSTAR to various experimental images with multiple objects, which present different challenges (Fig. 3f–j). We highlight that in all cases we train LodeSTAR on a single image (shown in the insets in the lower left corners in Fig. 3f–j) and then apply it to the whole image. In all cases, the objects are dividing over time, resulting in a large range of densities and morphologies that LodeSTAR needs to handle. Videos visualizing the analyzed sequences can be found for each case in the supplementary material (Supplementary movies 2–6).

First, we consider a dense sample (Fig. 3f) of mouse stem-cells, demontrating LodeSTAR's ability to identify the positions of very densely packed cells. Second, we consider a sample of human hepatocarcinoma-derived cells, where the cells vary highly both in morphology and in intensity (Fig. 3g). Finally, we consider a doubly challenging sample of pancreatic stem cells (Fig. 3h), where the cells are both densely packed and highly variable in morphology.

To compare the performance of LodeSTAR with other self-supervised methods, we construct an evaluation method. This method first compares predicted positions with publicly available annotations for the first three datasets[12]. It marks cells as found if a predicted position overlaps with the segmentation of the cell. Using this method, we evaluate the F1-score of LodeSTAR, as well as those of four other self-supervised object detection methods: SoCo[18], FSDet[21], InstanceLoc[20], and DETReg[19]. See Methods, "Object detection comparison", for more details.

Table 2 summarizes the results of the comparison. We find that LodeSTAR achieves results far superior to the alternative methods. We can also pinpoint the reasons for failure of the alternative methods. For the dataset in Fig. 3f, we find that they all perform well when the number of cells is low, but fail when the number of cells increase above some critical threshold. For the dataset in Fig. 3g, we find that they struggle at finding both the small and the large cells simultaneously given the small training sets. For the dataset in Fig. 3h, we find that they struggle to generalize at all beyond the training set, most likely because of the variability of the cells' morphology.

We also compare LodeSTAR to published results in the cell-tracking challenge[12]. We measure the DET* metric, where the * indicates

that it is a version of the DET score[26] that supports object detection methods that do not segment the objects (see Methods, "Object detection comparison"). We find DET*-scores of 0.989, 0.952 and 0.936 respectively; all of which are comparable to the top scores on the official benchmark[27].

LodeSTAR achieves these results despite being trained on more than 1000 times less data than the published methods. Although the comparison to the published methods cannot be made exactly since the official test set is not published (and the possibility of discrepancies between DET and DET*), these results demonstrate that LodeSTAR is at least comparable to state-of-the-art supervised object detection methods.

Finally, we consider some more complex objects with significant internal structure (Fig. 3i, j, see also Methods, "Plankton preparation and imaging"), namely some *Noctiluca scintillans*, large (400–1500 μm) single-celled dinoflagellate plankton. Since these planktons have a complex internal structure, LodeSTAR does not necessarily find their geometrical center, but it nonetheless consistently identifies some specific feature of their internal structure. Furthermore, we can expect this feature to depend on the details of the neural network and its training. In fact, when LodeSTAR is trained on the inset of Fig. 3i (150 px × 150 px), it learns to consistently identify the region where the tentacle attach and organelle aggregate. In contrast, when we downsample the images by a factor of 3 so that the training is made on the 50 px × 50 px figure shown in the inset of Fig. 3j, Lode-STAR consistently finds the plankton as a whole. In this way, it is possible to tune the scale of the detection performed by LodeSTAR.

### Exploiting additional symmetries

LodeSTAR can exploit additional symmetries to extend the range of object properties that it can measure. Holography is a prime example of an imaging modality ripe with additional symmetries[28]. Unlike ordinary brightfield microscopy where only the intensity of the incoming wave is imaged, holography provides access to the entire complex electromagnetic field. Consequently, it is possible to infer quantitative information about the imaged objects by measuring and manipulating the Fourier representation of the image. As we will see in the following sections, when analyzing holographic microscopy images, we can exploit the Fourier propagation symmetry to detect the axial position and the signal strength scaling symmetry to determine the polarizability of imaged particles. Implementation-wise, additional symmetries are encoded as additional channels of the intermediate feature-map, and are trained using their own set of equivariances in the same manner as for the in-plane position we have discussed in detail until now.

### 3D detection exploiting Fourier propagation symmetry

A natural extension of two-dimensional detection is to incorporate the third axial dimension, normal to the imaging plane. This can give crucial insight into the full volume dispersion of objects, as well as provide more data to calculate statistical measures about the objects' motility, such as their diffusion. An example of (the imaginary part) of an holographic image of a particle (a 228 nm-radius polystyrene sphere) is shown by the top slice ($z = 0$ μm) in Fig. 4a. The holographic image can be propagated to different planes (i.e., different axial positions from the focal plane) by employing Fourier transforms, as shown by the other slices in Fig. 4a. This provides an equivariance that LodeSTAR can learn, similar to the equivariances in the plane. By training on the image in the top slice of Fig. 4a, LodeSTAR learns to locate the polystyrene spheres in 3D space, as shown in Fig. 4b, where the measured vertical position is visualized as the distance above the image. Note that the refractive indices of the medium ($n_{medium}$) and the immersion oil ($n_{oil}$) change the apparent axial position of the particle[29]. We correct this by multiplying the measured axial position by a factor $n_{oil}/n_{medium} = 1.128$[29]. The particles were both detected and located in

3D space using LodeSTAR. Using a synthetic dataset replicating the experimental conditions, we find that LodeSTAR achieves an F1-score of 0.99 (see Fig. 5a for details).

We compare the vertical position predicted by LodeSTAR with that predicted by a traditional focusing approach where the vertical position is found by iteratively refocusing a region near a detection until a focusing criterion is met[10]. As shown in Fig. 4c, the two methods are in very good agreement. Moreover, we calculate the standard deviation of the error of the two methods based on the covariance estimate proposed in Ref. 30: LodeSTAR outperforms the classical approach significantly, reaching $\sigma_z = 105$ nm compared to $\sigma_z = 231$ nm.

As a second verification, we go beyond the single image localization and compare the predicted diffusion constants of each particle trace (acquired from the detections using a simple linear sum assignment approach), calculated either from its in-plane or axial movement. Since the diffusion constant is a statistical measure of the thermal motion of the particle, the diffusion of individual particles in different directions may not agree, while the ensemble of all particles should. As such, we compare the distribution of measured diffusion constants in Fig. 4d for in-plane movement and axial motion. We find that both in-plane diffusion and axial diffusion are distributed similarly, and that both show a peak at the expected diffusion of $D_{SE} = 0.97\ \mu m^2 s^{-1}$. Moreover, we find that the distributions closely agree with the expected distribution of diffusion constants obtained by simulating $10^4$ traces.

## Particle polarizability exploiting signal strength symmetry

The holographic image of a particle carries information not only about the particle position, but also about its morphology and composition. For example, the real part of the polarizability of an object (which henceforth will be referred to as just the polarizability for convenience) is proportional to the integrated phase acquired by the light passing through the particle, which is particularly relevant for biological objects, where refractive index and density are strongly correlated. In other words, for biological materials, it is possible to directly translate the polarizability of a particle to its dry mass[31]. The integrated phase, in turn, increases roughly proportional to the amplitude of the scattered light for small non-absorbing particles. Thus, there is an equivariance between the polarizability and the scale of the signal, which LodeSTAR can learn.

Figure 6a shows the real and imaginary parts of a simulated holographic image of a particle with radius 228 nm and refractive index 1.5. By multiplying these images by a factor (Fig. 6b), we can alter the signal scale and, therefore, the polarizability of the imaged particle. LodeSTAR is trained to estimate the logarithmic difference between the scale factors, in addition to the particle position. We remark that this equivariance does not constrain the absolute scale of the polarizability, which thus needs to be calibrated against some observation of known polarizability.

In Fig. 6c, we evaluate the trained LodeSTAR on simulated particles, varying their radius and their refractive index. The mean absolute percentage error remains below 10% for the majority of the considered range, only increasing for very low signal observations where noise would corrupt most of the signal. This is comparable to the findings of other methods, and yields accurate determinations by averaging over several observations of the same particle[10,32].

Next, we validate LodeSTAR's ability to measure the polarizability of particles in experimental data of 228 nm and 150 nm polystyrene beads. First, we measure LodeSTAR's ability to detect particles using a synthetic replica of the optical setup. We find that LodeSTAR achieves an expected F1-score of over 0.95 (see Fig. 5). To capture the population distribution of polarizability, we additionally need to link detections of the same particle over time. In this way, each particle is weighed equally in the distribution. We link particles by minimizing a linear sum assignment problem, which was found to be sufficient for this data.

In Fig. 6d, we consider a bi-dispersed sample of 150 nm radius and 228 nm radius polystyrene particles imaged through a holographic microscope. LodeSTAR is trained on one observation from the 228 nm population and is subsequently used to predict the polarizability of all particles in the sample. By calibrating against the 228 nm population, LodeSTAR successfully identifies the 150 nm population with high accuracy.

Having verified that LodeSTAR can reliably determine the polarizability of particles, we consider a biological sample with 225 nm-radius green-fluorescent polystyrene beads that were incubated with human neuroblastoma cell from the SH-SY5Y cells-line (Fig. 6e, see Methods, "Human neuroblastoma cell sample preparation"). The particles are simultaneously imaged with sample-position-modulated holography and fluorescence (see Methods, "Holographic imaging"). This allows us to classify detections as polystyrene particles (orange markers, which are fluorescent) or biological aggregates (blue markers, which are not fluorescent). As can be seen in the zoomed-in region in Fig. 6e, the signal is extremely low compared to the background. As such, one can expect spurious detections. We filter these out by only considering observations that could be linked over time for at least 40 frames, disregarding observations that arose from random noise. We find that the majority of detections are co-located with the cells. Further, studying the time-series (Supplementary Movie 8) the particles move in unison with the cells, supporting the premise that some of the particles have been taken up by the cells. See also details in Methods, "Measuring particle polarizability".

Finally, we study the distribution of polarizability for the particles and the biological matter in Fig. 6e. We find a much broader distribution for the polystyrene particles than the bi-dispersed case, which is expected if some of the particles are measured inside of the cells, since the intracellular medium has a higher refractive index than the surrounding medium (1.38 compared to 1.33[33]) and consequently yields slightly lower polarizability of the particle. Moreover, particles inside the cells are likely to be coated with biological material, further broadening the distribution. The peak of the distribution aligns well with the expected polarizability of polystyrene inside the cell, given a cytoplasmic refractive index of 1.38[33]. However, a second peak near the polarizability of polystyrene in water suggests that a significant portion of the particles have not entered the cells, which agrees with other cell–particle uptake experiments[34].

The distribution of polarizability of biological particles is significantly narrower and peaks at just under $0.01\ \mu m^3$. It should be noted that the detection of objects with polarizability lower than $0.006\ \mu m^3$ is not reliable due to the low signal, which means that the lower end of the distribution should be analyzed with some caution. Regardless, the two distributions are clearly distinct, indicating that they represent two separate physical properties of the sample.

## Discussion

We have developed a method, named LodeSTAR, that exploits the inherent symmetries of a problem to enable label-free training of neural networks using tiny datasets. We have demonstrated this capability by training neural networks to detect objects in a broad range of simulated and experimental scenarios. Moreover, we have shown that, in holography, LodeSTAR can quantitatively measure the objects in terms of their axial position and their polarizability. The software together with the source code and all the examples in this work are made publicly available through the DeepTrack 2.1 GitHub repository[35].

Compared to traditional approaches, we are able to achieve a better detection performance in terms of sub-pixel accuracy, while generalizing to more arbitrary morphologies. Moreover, unlike established methods for object detection using deep learning, which commonly require thousands of annotated images for training[2], LodeSTAR successfully detects the objects in difficult experimental

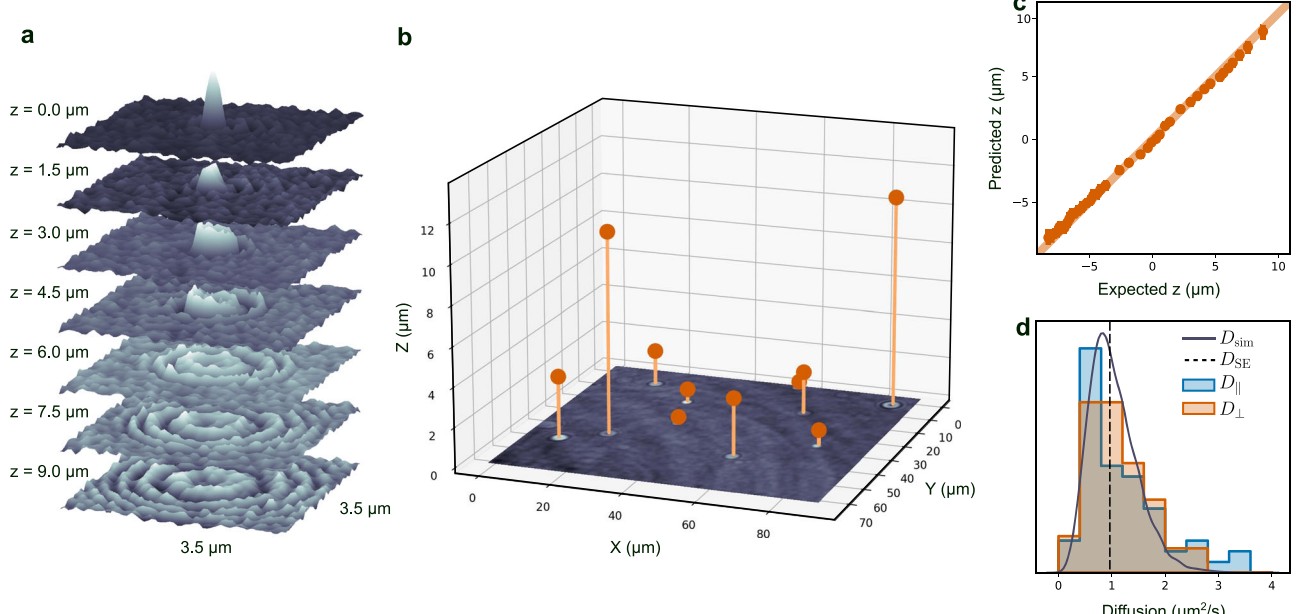

**Fig. 4 | LodeSTAR measurement of 3D positions exploiting Fourier propagation symmetry. a** Imaginary part of the hologram of a 228 nm radius polystyrene particle numerically Fourier-propagated to different axial distances from the focal plane. **b** LodeSTAR exploits this Fourier propagation symmetry to learn how to locate particles in three dimensions (orange positions). **c** The vertical position estimated by LodeSTAR agrees well with the expected position acquired using a traditional approach described in ref. 10. See also Supplementary Movie 7. **d** The distributions of the in-plane $xy$-diffusion (blue histogram) and the axial $z$-diffusion (orange histogram) of the particles show a strong peak at at the expected diffusion (dashed black line), and agree well with the expected theoretical distribution obtained by calculating the diffusion constant of $10^4$ synthetic traces (solid black line).

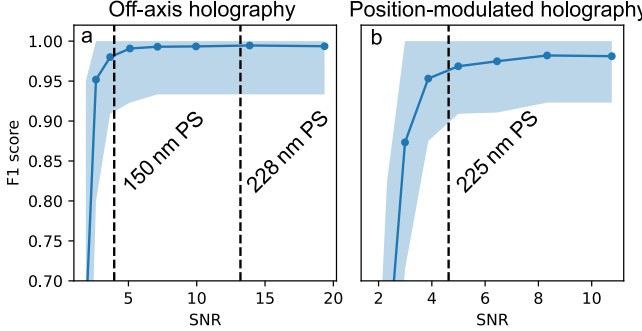

**Fig. 5 | Detection accuracy of LodeSTAR on holographic data.** The detection F1 score as a function of the signal-to-noise ratio (calculated as $\max(|I|)/\sqrt{\sigma_{Re}^2 + \sigma_{Im}^2}$, where $I$ is the field, and $\sigma_{Re}, \sigma_{Im}$ are the standard deviations of the real and imaginary parts of the field), on synthetic data. **a** F1 score (shaded region is the 0.95-th quantile) of LodeSTAR on the standard holographic setup. For the two particle sizes used, the F1 score is around 0.98. **b** F1 score (shaded region is the 0.95-th quantile) of LodeSTAR on the sample-position-modulated holographic setup. For the particle size used, the F1 score is around 0.96.

data using just a single datapoint for training. On top of this, LodeSTAR can be trained on an ordinary computer with no hardware acceleration (e.g., graphics processing units) within a few minutes.

As a novel approach to an established problem, we see several future lines of inquiry. Foremost, we expect the existence of many additional symmetries not considered in this work, particularly symmetries in the experimental design (such as the symmetries that allowed us the measure the axial position and polarizability of nanoparticles). We also expect that incorporating techniques from other areas of deep learning to be fruitful; for example, using active learning to optimize performance from low amounts of human supervision. Regarding the development of the technique itself, we consider the development of techniques to further improve the specificity of the

model, such as leveraging negative labels to indicate what not to detect. Finally, LodeSTAR may be used for segmentation, similarly to the results of the DINO (self-distillation with no labels) algorithm[36], leveraging the fact that only regions within the object are informative to its position.

The comparably low barrier of entry permits rapid creation of a custom detection method, requiring little-to-no expertize from the user. In addition, by side-stepping the need for synthetic data, Lode-STAR opens up the possibility to train high-quality models to analyze data that is difficult to reproduce synthetically, without relying on fallible human annotation.

## Methods
### Theory of geometric self-distillation
This section shows theoretically that LodeSTAR locates objects correctly. First, we consider the case of an arbitrary, constant object with no noise. Then, we show the special case of a symmetrical object. Finally, we consider an imperfect estimator. We will describe the theory to find the position of the object, but the same arguments easily extend to other properties for which a symmetry is available (e.g., mass, size, orientation).

Let $X$ be the set of possible images of an object, $Y$ be the set of possible positions of the object in the image, $f$ be the ground-truth function that returns the object position and $h$ be a neural network, both mapping $X \to Y$, and $G$ be the Euclidean group consisting of translations, rotations and reflections, acting on both $X$ and $Y$.

Given a single image $x_0 \in X$, we can define the subset $X^{x_0}$, which are all the elements of $X$ reachable by acting on $x_0$ with $G$. $f$ is, by definition, equivariant to $G$ on the subset $X^{x_0}$.

Now, additionally assume that (Assumption 1), $h$ is trained to be equivariant to $G$ on the subset $X^{x_0}$, i.e., $h(gx) = gh(x) \, \forall \, g \in G$ and $\forall x \in X^{x_0}$. Next, we defined the error **c** as the error $f(x_0) - h(x_0)$ for the input image $x_0$. Now, for any other $x' \in X^{x_0}$, we know that $x' = g'x_0$ for some $g' \in G$. Let us factor $g'$ into $g'_r g'_t$, where $g'_t$ is the translational part

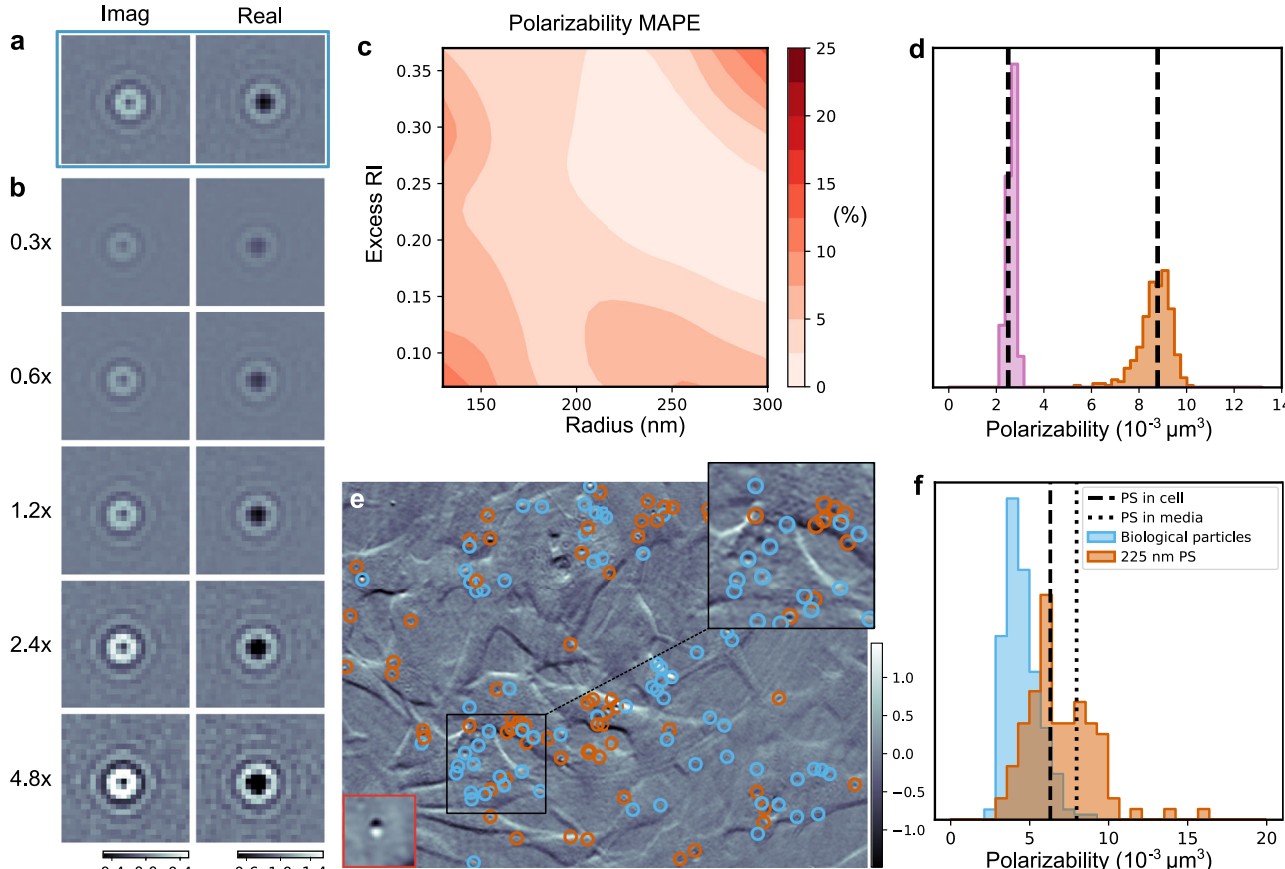

**Fig. 6 | LodeSTAR measurement of particle polarizability exploiting signal strength symmetry. a** Real and imaginary part of a simulated holographic image of a sphere (radius 228 nm, refractive index 1.58), (**b**) their versions with numerically rescaled signal strengths. **c** Despite being trained on a single particle (radius 228 nm, refractive index 1.58), the mean absolute percentage error (MAPE) of the predicted polarizability remains below 10% for a wide range of particle sizes and refractive indices. **d** In an experimental bi-dispersed sample, LodeSTAR accurately estimates the polarizability of the 150 nm population of polystyrene particles (pink histogram), even though it is trained on an image from the 228 nm population (orange histogram). **e** The imaginary part of a position-modulated holography image of fluorescent polystyrene particles (radius 225 nm) suspended inside and

around SH-SY5Y human neuroblastoma cells imaged through an off-axis holography microscope. LodeSTAR, trained on a single image (bottom left), learns to detect and measure the fluorescent particles (orange markers) as well as the non-fluorescent intracellular particles (blue markers). See also Supplementary Movie 8. **f** The distributions of the measured polarizability of the particles and the biological particles are drawn from two distinct distributions, indicating that we successfully separate the added polystyrene particles from the biological particles. The peak of the distribution matches the expected polarizability of polystyrene inside of cells (dashed line), a less prominent peak near the expected polarizability of polystyrene outside of cells (dotted line).

of the transformation, and $g'_r$ is the non-translational remainder. Finally, we define $h$ as "consistent" if $f(x') - h(x') = g'_r \mathbf{c}$ (i.e., the offset is independent of the translation and transforms correctly with $g'_r$). We can easily show this to hold by substitution:

$$f(x') - h(x') = f(g'x_0) - h(g'x_0) \tag{1}$$

$$= g'_r g'_t f(x_0) - g'_r g'_t h(x_0) \tag{2}$$

$$= g'_r f(x_0) - g'_r h(x_0) \tag{3}$$

$$= g'_r [f(x_0) - h(x_0)] \tag{4}$$

$$= g'_r [f(x_0) - f(x_0) + \mathbf{c}] \tag{5}$$

$$= g'_r \mathbf{c} \tag{6}$$

Note that $g'_t$ disappears in Eq. (3) since it equals to adding and subtracting a constant vector. In Eq. (4), we use the fact that $g'_r$ is linear.

This is sufficient to show that LodeSTAR, trained to be perfectly roto-translationally equivariant to some input image $x_0$, learns to detect identical objects consistently. However, for two different subsets $X^{x_0}$ and $X^{x_1}$, $\mathbf{c}$ may have different values. Consequently, internal consistency is not guaranteed between these two sets. If an action $g^*$ can be defined that can transform some element of either set to the other, then for the combined set $G \cup g^*$, $X^{x_0} \equiv X^{x_1}$ by definition. LodeSTAR is then guaranteed to be internally consistent for both subsets.

Now we demonstrate what happens if the images have some symmetry with respect to the group $G$. To do this, we assume that (Assumption 2) there exists a transformation $g \in G$ that is not a trivial transformation, such that $gx = x$ for some $x \in X^{x_0}$. This is known as a symmetry.

We know that $f(x') - h(x') = g'_r \mathbf{c}$. Now, due to Assumption 2, we know that there exists some non-trivial $g^*$ such that $g^*x = x$. Consequently,

$$g'_r \mathbf{c} = f(x') - h(x') \tag{7}$$

$$= f(g^*x') - h(g^*x') \tag{8}$$

$$= g^* f(x') - g^* h(x') \tag{9}$$

$$= g^* g_t^* f(x') - g^* g_t^* h(x') \tag{10}$$

$$= g_r^* f(x') - g_r^* h(x') \tag{11}$$

$$= g_r^* [f(x') - h(x'))] \tag{12}$$

$$= g^* g_r' \mathbf{c} \tag{13}$$

In other words, we have constrained $\mathbf{c}$ such that $g_r' \mathbf{c} = g_r^* g_r' \mathbf{c}$ holds. How this constraint manifests depends on the transformation $g_r^*$. If $g_r^*$ is equivalent to a reflection along some line, then $g_r' \mathbf{c}$ must be parallel to that line. This is equivalent to saying that the offset between $f$ and $h$ in the direction normal to a reflection symmetry must be 0. If $g_r^*$ is some rotation about a point, then $\mathbf{c}$ must be zero, since $y = g_{\mathrm{rotation}} y$ can only hold if $g_{\mathrm{rotation}}$ is a trivial rotational transformation (i.e., rotating an integer multiple of $2\pi$), or if $y = \mathbf{0}$.

We find that any rotational symmetry necessarily means that $f = h$ on $X^{x_0}$. In fact, we have done so without defining $f$ more exactly than being equivariant to $G$. Further, we can show that $f$ and $h$ necessarily find the center of rotational symmetry. Consider an $x$ that is symmetric to a purely rotational transformation, $g_{\mathrm{rotation}} x = x$. Then $h(x) = h(g_{\mathrm{rotation}} x) = g_{\mathrm{rotation}} h(x)$. This can only be true if $h(x)$ is the center of rotation, for the same reason that $\mathbf{c}$ had to be 0.

In reality, Assumption 1 will never hold exactly. LodeSTAR will not be perfectly equivariant to the Euclidean group. As such, it is important to verify that a small deviation from equivariance results in a small error in predicted position. We assume that $h(gx) = gh(x) + \mathbf{e}_{x,g}$ for some small $\mathbf{e}_{x,g}$. We recalculate Eqs. (1)–(6) with this new assumption.

$$f(x') - h(x') = f(g' x_0) - h(g' x_0) \tag{14}$$

$$= g_r' g_t' f(x_0) - g_r' g_t' h(x_0) - \epsilon_{x',g'} \tag{15}$$

$$= g_r' f(x_0) - g_r' h(x_0) - \mathbf{e}_{x',g'} \tag{16}$$

$$= g_r' [f(x_0) - h(x_0)] - \mathbf{e}_{x',g'} \tag{17}$$

$$= g_r' [f(x_0) - f(x_0) + \mathbf{c}] - \mathbf{e}_{x',g'} \tag{18}$$

$$= g_r' \mathbf{c} - \mathbf{e}_{x',g'} \tag{19}$$

Then, substituting this into Eqs. (7)–(13), we find that

$$g_r' \mathbf{c} = f(x') - h(x') + \mathbf{e}_{x',g'} \tag{20}$$

$$= f(g^* x') - h(g^* x') + \mathbf{e}_{x',g'} \tag{21}$$

$$= g^* f(x') - g^* h(x') + \mathbf{e}_{x',g'} \tag{22}$$

$$= g^* g_t^* f(x') - g^* g_t^* h(x') + \mathbf{e}_{x',g'} \tag{23}$$

$$= g_r^* f(x') - g_r^* h(x') + \mathbf{e}_{x',g'} \tag{24}$$

$$= g_r^* [f(x') - h(x')] + \mathbf{e}_{x',g'} \tag{25}$$

$$= g_r^* \left[ f(x') - h(x') + \mathbf{e}_{x',g'} - \mathbf{e}_{x',g'} \right] + \mathbf{e}_{x',g'} \tag{26}$$

$$= g_r^* g_r' \mathbf{c} + \mathbf{e}_{x',g'} - g_r^* \mathbf{e}_{x',g'} \tag{27}$$

Note that in Eq. (22), we can omit the $\mathbf{e}_{x',g^*}$ since $x = g^* x$, so $\mathbf{e}_{x',g^*}$ must be 0. We see that $g_r' \mathbf{c} - g_r^* g_r' \mathbf{c} = \mathbf{e}_{x',g'} - g_r^* \mathbf{e}_{x',g'}$. Remembering that $g_r^*$ acts on $R^2$ like a matrix, we see that $(\mathbf{I} - g_r^*) g_r' \mathbf{c} = (\mathbf{I} - g_r^*) \mathbf{e}_{x',g'}$. If $(\mathbf{I} - g_r^*)$ is invertible (as is the case for rotations), we find that $g_r' \mathbf{c} = \mathbf{e}_{x',g'}$. In other words, the difference between $f$ and $h$ is equal to the error associated with imperfect equivariance. If this error is small, then $g_r' \mathbf{c}$ is also small.

If $(\mathbf{I} - g_r^*)$ is not invertible (as is the case for reflections), we need to consider its eigenvalues. In particular (for convenience), its left eigenvalues. For any left eigenvector $\mathbf{v}_\lambda$ with a eigenvalue $\lambda$, we find that

$$(\mathbf{I} - g_r^*) g_r' \mathbf{c} = (\mathbf{I} - g_r^*) \mathbf{e}_{x',g'} \tag{28}$$

$$\mathbf{v}_\lambda^T (\mathbf{I} - g_r^*) g_r' \mathbf{c} = \mathbf{v}_\lambda^T (\mathbf{I} - g_r^*) \mathbf{e}_{x',g'} \tag{29}$$

$$\lambda \mathbf{v}_\lambda^T g_r' \mathbf{c} = \lambda \mathbf{v}_\lambda^T \mathbf{e}_{x',g'} \tag{30}$$

If $\lambda = 0$, this reduces to $0 = 0$, which does not restrict $g_r' \mathbf{c}$. If $\lambda \neq 0$, then we see that the projection of $\mathbf{c}$ onto $\mathbf{v}_\lambda$ is equal to the projection of $\mathbf{e}_{x',g'}$ onto $\mathbf{v}_\lambda$. The error in this direction must then be small if $\mathbf{e}_{x',g'}$ is small.

Finally, it should be noted that to generalize this argument to other properties one needs to be careful of trivial solutions. For example, in the absence of a translation-like transformation, there will often be a trivial solution at $h \equiv \mathbf{0}$. Symmetries to $G$ can only exist if the center of symmetry is at origin, whereby $h \equiv \mathbf{0}$ is a correct, yet woefully uninformative solution. As a rule of thumb, a group consisting of one translation-like transformation is sufficient for LodeSTAR to be internally consistent, while a group that additionally has some linear transformation (with a corresponding symmetry) is sufficient to ensure that there is only one optimal solution.

## Neural network architecture

The neural network consists of three $3 \times 3 \times 32$ convolutional layers with ReLU activation, followed by a $2 \times 2$ max-pooling layer, followed by eight $3 \times 3 \times 32$ convolutional layers with ReLU activation, and finally by a single $1 \times 1 \times 3$ convolutional layer with no activation. The architecture is designed as a balance between retaining spatial information necessary for high-precision localization (by limiting the number of pooling layers), and acquiring a big enough receptive field to most purposes. Splitting the output channels into two-dimensional tensors $\Delta x$, $\Delta y$ and $\rho$ respectively, LodeSTAR first calculates

$$x_{i,j} = \Delta x_{i,j} + ik - \frac{N}{2}, \tag{31}$$

$$y_{i,j} = \Delta y_{i,j} + jk - \frac{M}{2}, \tag{32}$$

where $N$ is the size of the output tensor along the first dimension, $M$ is the size of the output along the second dimension, and $k$ is a scale factor relating the size of the input to that of the output ($k = 2$ for the proposed architecture). Any additional channels (such as for 3D-positioning or polarizability estimation) are used without any modification.

The weigth map $\rho$ is normalized to $w$ so that

$$w_{i,j} = S(\rho_{i,j}), \tag{33}$$

where $S(\cdot)$ is the sigmoid function. Here, it is used to constrain individual elements between 0 and 1, which makes choosing a detection threshold for multi-object detection easier. Thus, the tensors $x$, $y$, and $w$ are the final outputs of the neural network.

Ignoring edge effects (which is reasonable if $w$ is mostly 0 with a few non-zero regions), we find that the weighted averages,

$$\bar{x} = \frac{\sum_{i,j=1}^{N,M} x_{i,j} w_{i,j}}{\sum_{i,j=1}^{N,M} w_{i,j}}, \tag{34}$$

$$\bar{y} = \frac{\sum_{i,j=1}^{N,M} y_{i,j} w_{i,j}}{\sum_{i,j=1}^{N,M} w_{i,j}}, \tag{35}$$

are translation equivariant.

### Neural network training

LodeSTAR is trained using images of individual objects in isolation. For each mini-batch, one such image is chosen and randomly transformed using a transformation function $\tau_{\theta_k}(\cdot)$, where $\theta_k$ are randomly sampled parameters for each sample in the mini-batch. For object detection, $\tau_\phi$ is a roto-translational transformation. Several transformed views of the original image (8 for all experiments in this paper) are combined into a single mini-batch.

Let $\chi_k$ and $\rho_k$ be the predicted feature channels and the weight channel for the $k$-th sample in the mini-batch. Note that $\chi_k$ includes the calculation of $x$ and $y$ as described in the previous section. Here, $\rho_k$ is normalized as:

$$w_{k,i,j} = \frac{\epsilon + D_{0.01}[S(\rho_{k,i,j})]}{MN\epsilon + \sum_{n,m=1}^{N,M} D_{0.01}[S(\rho_{k,n,m})]}, \tag{36}$$

where $\epsilon$ is some small value $\left(10^{-6}\right)$, $D_{0.01}[\cdot]$ is a dropout with a dropout-rate of 1%, and $k, i, j$ are the batch and spatial indices. The dropout helps avoiding the solutions where a single element is large and the rest are small, increasing the robustness of the network. The $\epsilon$ effectively assigns a minimum weight to each output pixel. Since the network strives to minimize the weight of non-informative regions, it is incentivized by the $\epsilon$ to maximize the term $\sum_{n,m=1}^{N,M} D_{0.01}[S(\rho_{k,n,m})]$, which is maximized by predicting large values at informative regions. As such, $\epsilon$ forces the network to utilize the full span of the sigmoid function.

Then, we compute the weighted average of each channel $c$,

$$\bar{\chi}_k^c = \frac{\sum_{i,j=1}^{N,M} \chi_{k,i,j}^c w_{i,j}}{\sum_{i,j=1}^{N,M} w_{i,j}}, \tag{37}$$

which are subsequently transformed as

$$\bar{\chi}_k'^c = \tau_{\theta_k}^{-1}(\bar{\chi}_k^c). \tag{38}$$

We employ two loss functions during training. The first is a loss between the inversely transformed averaged predictions and their batch-wise mean, calculated as

$$\mathcal{L}_a^c = \sum_{k=1}^{K} |\bar{\chi}_k'^c - \frac{\sum_i^K \bar{\chi}_i'^c}{K}|, \tag{39}$$

where $k$ and $i$ are batch indices, and $K$ is the number of samples in a mini-batch.

The second is an internal consistency loss, which is calculated as

$$\mathcal{L}_b^c = \sum_{k,i,j=1}^{K,N,M} |\chi_{k,i,j}^c - \bar{\chi}_k| w_{k,i,j}, \tag{40}$$

This second auxiliary loss ensures that the prediction is internally consistent, and is what causes the clustering of the prediction. Note that if the model predicts $\rho = 1$ for one position and 0 everywhere else, this metric would be 0. As such, it encourages a tight spatial weight distribution, which is useful for separating closely packed objects. However, the dropout in the weight during training ensures that it does not collapse to a single pixel, which would remove eny clustering of the output. Without this loss, LodeSTAR would occasionally ignore the feature maps, and instead tune the weights such that only the average is correct.

In all cases, LodeSTAR was trained using the Adam optimizer[37], with a learning rate of 0.001. Unless otherwise stated, the model was trained using 5000 mini-batches of 8 samples. The exception is for the 3D detection described in Fig. 4, where the model is trained on 15,000 mini-batches.

### Evaluating the Cramer-Rao lower bound

For a pure Poisson process, the Fisher information matrix is calculated as

$$\mathbf{I}(\theta) = \sum_{k=1}^{K} \left[ \left(\frac{\partial \nu_{\theta,k}}{\partial \theta}\right)^T \left(\frac{\partial \nu_{\theta,k}}{\partial \theta}\right) \frac{1}{\nu_{\theta,k}} \right], \tag{41}$$

where $k$ enumerates $K$ pixels in the image, $\theta$ is the parameter vector, and $\nu_{\theta,k}$ is the expected number of photons for a given pixel and detector[22]. Since the images are all synthetic, $\nu_{\theta,k}$ is known exactly.

We considered the $x$ and $y$ position of the object, as well as the orientation $\alpha$ as the parameters. To calculate the partial derivatives, we simulated $\nu_{\theta,k}$ at 10 times higher resolution than the test images and further interpolated the spacing between the pixels. Finally, the bounds on the variance of an estimator are calculated as the diagonal elements of the inverted information matrix, i.e., $[\delta_x^2, \delta_y^2, \delta_\alpha^2] = \mathbf{I}^{-1}(\theta)$. The bound on the root mean squared error is finally calculated as $\sqrt{\frac{\delta_x^2 + \delta_y^2}{2}}$.

### Particle detection criteria

Particles are detected by finding local maxima in a score map using the function h_maxima of the Python module skimage. The score map is based on two separate metrics. The first is the weight map $w$. The second is a clustering metric calculated as

$$b_{ij}^{-1} = \sum_{c=1}^{C} \left[ \left( \sum_{j'=j-1}^{j+1} \sum_{i'=i-1}^{i+1} \frac{\chi_{ij'}^{c\,2}}{9^2} \right) - \left( \sum_{j'=j-1}^{j+1} \sum_{i'=i-1}^{i+1} \frac{\chi_{ij'}^c}{9} \right)^2 \right], \tag{42}$$

where $\chi^c$ are the feature maps produced by the network, and C are the number of feature channels. This can be likened to convolving the feature map with a variance kernel. The two metrics are combined geometrically as $w^\alpha b^\beta$, where $\alpha$ and $\beta$ can be tuned to optimize performance. In the cases presented in this paper, we consistently use $\beta = 1 - \alpha$. For the cases in the chapter "Validation with experimental data", we use, in order, $\alpha = 0.2$, $\alpha = 1$, $\alpha = 1$, $\alpha = 1$. A high $\alpha$ is used for the last few examples since the cells are not very similar, which typically leads to poorer clustering performance. As seen in Table 1, the neural network's ability to cluster its predictions is highly dependent on the similarity between the evaluation data and the training data. As such, it can be expected to be a poorer detection metric if objects vary highly in shape. For the remaining cases, we use $\alpha = 0.1$. Local maxima in this product are taken as observation if they

are larger than some threshold, which in turn is chosen as a quantile of all scores. The quantile can be chosen a priori based on the density of the experimental data, but it can commonly be taken as the 99th percentile.

### Object detection comparison

Each evaluated dataset consists of two sequences of images. One sequence was reserved for training and the other for testing. The testing sequence was chosen as the one with the most cells. The alternative methods (SoCo[18], FSDet[21], InstanceLoc[20], and DETReg[19]) were initialized from weights pretrained on the CoCo dataset[38] or the ImageNet[39] dataset, as provided by their respective author. From the training dataset. LodeSTAR was trained on one crop of one cell, while the alternative methods were fine-tuned on one full image per dataset, each of which contained two cells. LodeSTAR was trained for 30 epochs with a batch size of 8, while the alternative methods were trained for 5000 epochs with the batch size recommended by the author. To avoid overfitting, the alternative models were evaluated on the test set every 10 epochs, and the final weights were chosen at the optimal end-point.

To evaluate the models, we define a method to match predicted cell positions with the ground truth annotations consisting of segmentation markers. If a single position prediction overlap with the segmentation of a cell, that prediction is considered a true positive. If multiple predictions overlap, each additional prediction is considered a false positive. Each prediction not overlapping with a segmentation is considered a false positive, and each segmentation with no corresponding prediction counts as a false negative. For methods that return a bounding box, the center of the box was taken as the predicted position.

Finally, we evaluate both the F1-score and the DET[*]. The F1-score is defined as $\frac{2TP}{2TP+FP+FN}$, where TP is the number of true positives, FP is the number of false positives, and FN is the number of false negatives. We count the number of true positives, false positives, and false negatives using the method described above. The base DET metric is defined as $1 - \frac{\min(D,D_0)}{D_0}$ where $D$ is the Acyclic Oriented Graph Matching measure for detection[26]. This measure determines that a ground truth marker was found if a predicted marker covers more than half the ground truth marker. For the DET[*] metric, we replaced this matching condition with the matching method used to calculate the F1 score, as described above.

### Plankton preparation and imaging

*Noctiluca scintillans* (SWE2020) was isolated from the Swedish west coast in November 2020. The culture was maintained in 16 °C, 26 psu, and 12:12 h light:dark cycles. The culture flasks were shaded by a screen to limit growth of the food organisms, *Dunaliella tertiolecta*. *Noctiluca* cultures were fed ad libitum. The planktons are imaged with an inline holographic microscope, illuminated with a LED source (Thorlabs M625L3) of center wavelength 632 nm (details in[40]). The images are recorded with a CMOS sensor (Thorlabs DCC1645C) placed at a distance of ≈ 1.5 mm from the sample well, at 10 frames per second and with an exposure time of 8 ms.

### Human neuroblastoma cell sample preparation

SH-SY5Y cells were grown in cell culture media (CCM) containing a 1:1 mixture of minimal essential medium (MEM) and nutrient mixture F-12 Ham supplemented with 10% heat-inactivated fetal bovine serum, 1% MEM nonessential amino acids, and 2 mM l-glutamine. The cells were detached (trypsin-EDTA 0.25%, 5 min) and passaged twice a week. The cells were tested and verified mycoplasma-free. Cells were plated 1 day prior to experiments in glass-bottomed culture dishes (MatTek; 25000 cells/14 mm glass region) for microscopy. Cells were washed 1× with serum-free CCM before exposure to 225 nm green-fluorescent polystyrene particles diluted in serum-

containing CCM with the addition of 1% Penicillin-Streptomycin. After a 4 h incubation at 37 °C 5% CO2, the cells were washed twice for 2 min with serum-free CCM, followed by addition of serum-containing CCM supplemented with 1% Penicillin-Streptomycin and 30 mM HEPES to buffer the medium. The cells were imaged at 37 °C using a OKOlab stage top incubator.

### Holographic imaging

The used monodisperse particles are and 0.15 μm (modal radius) polystyrene (Invitrogen) and 0.23 μm (modal radius, NIST-certified standard deviation ± 6.8 nm) polystyrene (Polysciences) (sizes verified using nanoparticle tracking analysis performed by NanoSight). Samples were imaged under flow in straight hydrophilized channels with a height of 20 μm and a width of 800 μm in chips made from Topas (COC, ChipShop). The images were captured using an off-axis holographic microscope[10], using a 633 nm HeNe laser (Thorlabs) and a Olympus 40 × 1.3 NA oil objective. The interference pattern was collected using a CCD camera (AlliedVision, ProSilica GX1920), at a frame-rate of 30 frames per second, and an exposure time of 2 ns–4 ms.

For the intracellular data, a 40×, 0.95 NA Objective (Nikon, CFI Plan Apokromat) objective was used instead. The particles consisted of 225 nm-radius green-fluorescent polystyrene spheres (PS-FluoGreen, microparticles Gmbh). The particles were diluted 5000 times in cell media from the stock solution concentration of 2.5 wt% to a concentration of 5.25 μg ml⁻¹. Further, to improve the data aquisition quality, the stage was oscillated at 1.1 μm roughly every 4 s and offset pairs of frames were subtracted from each other to mitigate noise from reflections (sample-position-modulated holography). The sample was imaged at 3 frames per second.

The fluorescence arm was illuminated using a 465 nm LED excitation (CoolLED), and imaged using a ORCA-Flash 4.0 V2.0 CMOS camera (Hamamatsu). We filtered the signal using a 491 nm dichromatic mirror (Chroma Technology Corporation), and separated the fluorescence channel from the holography channel using a long-pass dichroic mirror (605 nm), as well as a 525 ± 39 nm bandpass emission filter from Thorlabs.

### Measuring particle polarizability

Particles were detected in each frame using a neural network trained by LodeSTAR using a single observation. The observations were subsequently traced over time by linear sum assignment, with a distance threshold of 1 μm. The polarizability was averaged over measurements from each detection in a trace. The LodeSTAR-measured polarizability was calibrated against a known population as $\alpha = 3V \frac{n_p^2 - n_m^2}{n_p^2 + 2n_m^2}$, where $\alpha$ is the polarizability, $V$ is the volume of the particle, $n_p$ is the refractive index of the particle and $n_m$ is the refractive index of the medium[41]. Further, for the intracellular data, each detection was compared to a fluorescence channel. If the detection was within 400 nm of a fluorescence detection, it was considered a polystyrene particle. If >95% of detections in a trace were linked to a fluorescence detection, the trace was considered a trace of a polystyrene particle. If <5% of detections in a trace were linked to a fluorescence detection, the trace was considered a trace of an intracellular particle.

## Data availability

All in-house data is available for download through the respective example at the DeepTrack-2.1 GitHub repository[35]. The remaining data can be accessed from the Cell Tracking Challenge website[12].

## Code availability

All source code and examples are made publicly available at the DeepTrack-2.1 GitHub repository[35]. The version used in this study is archived in Zenodo with DOI 10.5281/zenodo.7175126.

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

## Acknowledgements

The authors would like to acknowledge funweding from the H2020 European Research Council (ERC) Starting Grant ComplexSwimmers (Grant No. 677511) (G.V.), the Horizon Europe ERC Consolidator Grant MAPEI (Grant No. 101001267) (G.V.), the Knut and Alice Wallenberg Foundation (Grant No. 2019.0079) (G.V.), and Vetenskapsrådet (grant numbers 2016-03523 (G.V.) and 2019-05071 (D.M.)).

## Author contributions

B.M. and G.V. conceived the method. B.M. designed, implemented and tested the method. J.P., F.S., D.M., and G.V. contributed to the development of the method. H.B. and E.S. collected and imaged the plankton data. E.O. collected the holographic data, using an experimental setup and software developed by E.O., B.M., D.M., and F.H. E.W. and E.K.E. provided the SH-SY5Y cell cultures and analyzed the polystyrene particle uptake. E.O. performed the cell measurements, within a methodological framework designed by E.O., F.H., D.M., E.W., and E.K.E., and experimentally realized by E.O. G.V. supervised the work. B.M., J.P., and G.V. drafted the paper and the illustrations. All authors revised the paper.

## Funding

## Competing interests

B.M., J.P., D.M., and G.V. are co-founders of the company IFLAI AB to provide AI-solution for microscopy. The other authors declare no competing financial or non-financial interests.
