## [Peer Review File · Nature Communications]

Single-shot self-supervised object detection in microscopyREVIEWER COMMENTS

Reviewer #1 (Remarks to the Author):

Summary:

The authors propose a method to detect particle centroid using an unlabeled image. They utilize the property that the positions of particle centroids are equivariant to Euclidean transformations to perform self-supervised learning on a single-particle image. If a translation-equivalent neural network is used, it can be directly extended to detect multiple particles without extra training. Some techniques such as auxiliary loss and epsilon to the weight map are introduced during the training phase to avoid training collapse.

Overall, I am very impressed with the promising performance achieved by this method using only a single unlabeled image for training. The extremely low training data requirement makes it highly affordable for practical applications. I believe it will be of great help to the analysis of small particles in the field of biology and medicine.

Pros:

Single unlabeled image tasks for bio etc. is very important and with big potential impact
The impressive results
Novel methods with solid experiments and details

Major concerns:

1. The word "tracking" is misleading in many places in the manuscript, including in the title. Particle tracking should include particles detection and particles connection at different times to get the individual trajectories. For example, "We have demonstrated this capability by training neural networks to track objects...". I believe the neural network only learns to detect, not track, the particles.

From the perspective of computer vision, the work focusing on Detection, rather than tracking.

2. The authors only mention how they connect two particles in different frames in the last section of the "Methods", titled "Measuring particle polarizability". The expression here is very unclear if the authors want to focus on particle tracking.

Minor:

1. The normalization factor W' is first mentioned in the "Neural Network Training" section and lacks sufficient explanation.

Typos:

1. Page 4, 3rd line: 'Theory of ...'. Double-quotes.
2. Page 13, 2nd equation of internal consistency loss: the last y_{ij} should be w_{ij} .
3. Page 14, Particle detection criteria: missing period after "we use $\alpha = 0.1$ ".

Reviewer #2 (Remarks to the Author):

This paper presents a novel method (LodeSTAR) for particle identification using machine learning that uses unsupervised learning with very small training sets. I find the work to be

compelling and the demonstrations of the method to be persuasive. However, I have some concerns about the overall presentation of the manuscript as well as a couple of questions. Therefore I would recommend publication of this article after revisions. My feedback is summarized below:

1) This may be a requirement of the journal, in which case this comment can be ignored, but I found the placement of the Methods section at the end of the paper to be very frustrating. It should be placed in between the Introduction and Results sections, so that the reader can understand how the procedure works and why before being shown the results.

2) The authors present the algorithm as a "tracking" algorithm, but I believe all the demonstrations of the algorithm are for object identification rather than explicit tracking. I.e., objects are identified in single frames, but it was not demonstrated that object A in one frame can be successfully associated with object A in the second frame, and so on for object B, object C, and so forth. Without this demonstration, the method is an identification technique rather than tracking. Object identification is still very valuable, but this distinction is important.

3) Related to the previous point, LodeSTAR is based on equivariance of certain properties, primarily invariance under rotation and translation. These are sound for object identification, as demonstrated, but I question their appropriateness for tracking purposes. For tracking rigid particles, these equivariances hold under ideal conditions, but they may not in real situations. The same object visualized twice, especially after motion, may appear differently depending upon the optical diagnostic used. This is true for laser-based scattering, for instance, where a particle's appearance is highly dependent upon its location and orientation relative to the incident light. For non-rigid objects such as cells, however, does the deformation not become important? Between frames in a sequence, it is possible that a non-rigid object could deform (i.e., change shape) in addition to translation and rotation. What effect would this have on LodeSTAR's ability to track that object, if indeed the algorithm is capable of tracking between frames rather than just object identification?

Reviewer #3 (Remarks to the Author):

The authors present a method that belongs to the category of self-supervised techniques (more precisely self-distillation techniques) capable of localizing particles or small simple-shape objects in bioimage analysis without the necessity of creating labels for training. The idea of bringing this type of techniques (that are currently rather intensively researched in the general computer vision community) to bioimage analysis (where supervised techniques still dominate) is surely useful and logical. However, there are a number of serious issues that need to be addressed to make the paper acceptable for publishing.

Major issues:

1) Incorrect terminology including the title and abstract that do not correspond to the paper content at all.

1A) The authors speak about tracking in the whole paper but there is no tracking in the paper, rather there is just detection that precedes tracking. Tracking is about building tracks (or even lineage trees for dividing cells), i.e. interconnecting detections between frames (along the time axis). The authors use videos (time-lapse image sequences) but perform just detection per frame, no subsequent tracking. They even do not discuss any tracking papers /

methods (neither the famous Particle Tracking Challenge – not cited at all). Hence, tracking should be replaced with detection in the whole paper.

1B) The authors mostly speak about particles, sometimes objects. They denote also small objects like cells as particles. Cells are not particles. Particles are nearly point-like objects with few pixel diameters for which segmentation does not make sense. Hence, authors should speak about objects, not particles, starting from the title.

1C) The authors should admit right in the title and abstract that they do not speak about all objects but only bioimaging (microscopy) ones that are small (they work with few tens of pixels in diameter) and have rather simple roundish shapes (i.e., not tubular ones, not complex shapes with protrusions, etc.), homogenous interior (no complex texture inside) and, most importantly, are located on a homogenous background (e.g., not cells in tissues). A suitable title might be something like “Single-shot self-supervised small object detection in bioimaging” with a constraint explanation in the abstract.

2) Missing state-of-the-art.

The authors do not cite any relevant papers from the computer vision community on self-supervised (or cell-distillation) object detection (or even segmentation) approaches (with the exception of [22] in one sentence in the conclusion). There are many recent relevant papers, e.g.

<https://arxiv.org/abs/2110.14711> (survey with a nice overview Figure)

<https://arxiv.org/abs/2109.14279>

<https://www.biorxiv.org/content/10.1101/2021.01.07.425773v1>

<https://arxiv.org/abs/2010.12023>

3) Missing comparison of the developed method with state-of-the-art

The authors compare the developed method (in Fig.1g-k) with just two other rather old methods: trivial centroid approach ([12] from 1996) and radial symmetry centers ([13] from 2012). I do not think this is enough. Some representatives of modern state-of-the-art object detection methods should be added, e.g., self-distillative BYOL-based approaches such as SoCo (see the survey mentioned above). Even if the output of the modern method is a bounding box, it can be converted to the center output by computing the center of the bounding box. It would also be nice to compare the self-supervised approaches to the supervised ones. As nearly all the real (non-synthetic) datasets used in the paper are taken from the Cell Tracking Challenge [11] (Fig.2 f-h), it would be useful to compare the detection results with those listed in the current leaderboard for detection (DET measure of Cell Segmentation Benchmark). To measure DET, it would be necessary to replace center results with some circular markers (of reasonable size corresponding to average object size) to ensure overlap with at least half of the markers in [11] but should be doable. Currently, the performances on real datasets in Fig.2 are not compared to any other methods at all!

4) Problems with metrics.

The authors use RMSE for single simple synthetic object studies in Fig.1, which is fine, but then use TPR+FDR for real datasets, which is not compatible with other studies and causes the problem of not comparing the developed method performance to any other method (as mentioned above). Most other detection studies typically use F1-Score (or AP for probabilistic outputs of neural networks). If subsequent tracking is anticipated (which is the case of videos here), then recall is favored to precision like in [11] where DET measure (similar to F3-Score) is used. Anyway, the authors should use a measure compatible with other studies (or should measure TPR+FDR for other methods, which is more time-

consuming). Also for holography experiments (Figure 4e) some metric should be used – here the detection performance (for blue and orange markers) is not measured here at all.

5) Unclear TP definition.

The authors have not even described how they define TP (and consequently FP and FN) for the computation of TPR+FDR. Usually, some threshold distance is chosen to match the predicted center position with the ground truth center position but even in this case, it is necessary to solve ambiguities (like multiple predictions matching one ground truth object or vice versa). Also, in [11] markers are used instead of single positions, did the authors convert markers to positions?

6) 3D capabilities unclear.

It is not clear whether the method is applicable only in 2D or also in 3D. From the abstract, it seems 3D is addressed as well: “We also exploit additional symmetries to extend the measurable particle properties to the particle's vertical position”. However, the authors use 2D examples (and descriptions) throughout the paper with the exception of the 3D holography example. What about classical (non-interference) 3D images?

7) Unclear roto-translational symmetries.

Already in the abstract, the authors speak about “exploiting the inherent roto-translational symmetries of the data” as the basis of their method. Also, the very first section of Methods is devoted to symmetries as the basis of their method, here they define assumptions (1)-(4). However, objects in the experimental data often do not exhibit such symmetries, not even those objects used in the paper in Figure 2gh – they are not rotationally symmetric. For real world the assumptions (1)-(4) simply do not hold: (1) because h is not ideal (there is some error remaining after optimization), (2) because one cell is not just equal to rotated and translated another cell, (3)-(4) is admitted by the authors as not fulfilled for asymmetrical objects. So, there is an inconsistency between the theory behind the method and the properties of experimental objects. This inconsistency should be addressed somehow. If the method really works for “arbitrary shapes” (as mentioned on p.5), then it should be explained why.

8) The mathematical symbols and equations are not properly defined.

In many places in Methods, there are symbols in equations or text that are not defined or used interchangeably with sub-indices and without them, which makes it hard to understand the theory. Moreover, there are no accompanying figures visualizing the description (just a very poor network symbol in Figure 1c with no details).

Neural network architecture: undefined x , y , ρ , i , j , unclear what precisely is at the input, which weights are referred to (there are many weights in each network), why N is size without tensors and M with tensors, etc.

Neural network training: sums without bounds (not clear from which value to which the sum runs), undefined $W_{i,j}$ (used after sum), x^{ω} with missing sub-indices and missing bounds for summation, etc.

Particle detection criteria: undefined K (Number of clusters? How is it determined?), the sum over k should probably be applied to the whole remaining part, not just first parentheses (i.e., additional parentheses required), $\alpha = 1$ for three cases means $\beta = 0$, i.e. clustering not applied at all – deserves a comment.

Measuring particle polarizability: all symbols undefined and no citation. At least Clausius-Mossotti relation deserves a citation.

Minor issues:

1) There are some typos in the paper, e.g., "centriod" instead of "centroid" twice, "results of achieved with" (p.12), "f and h be functions h" (p.12), "in in" (p.12), "are and" (p.14).

Reviewer # 1

Summary:

The authors propose a method to detect particle centroid using an unlabeled image. They utilize the property that the positions of particle centroids are equivariant to Euclidean transformations to perform self-supervised learning on a single-particle image. If a translation-equivalent neural network is used, it can be directly extended to detect multiple particles without extra training. Some techniques such as auxiliary loss and epsilon to the weight map are introduced during the training phase to avoid training collapse.

Overall, I am very impressed with the promising performance achieved by this method using only a single unlabeled image for training. The extremely low training data requirement makes it highly affordable for practical applications. I believe it will be of great help to the analysis of small particles in the field of biology and medicine.

Pros:

Single unlabeled image tasks for bio etc. is very important and with big potential impact

The impressive results

Novel methods with solid experiments and details

We sincerely thank the reviewer for their praise and for recognizing the potential practical value of our method in microscopy.

Major concerns:

1. The word “tracking” is misleading in many places in the manuscript, including in the title. Particle tracking should include particles detection and particles connection at different times to get the individual trajectories. For example, “We have demonstrated this capability by training neural networks to track objects...”. I believe the neural network only learns to detect, not track, the particles. From the perspective of computer vision, the work focusing on Detection, rather than tracking

We agree with the reviewer that the word “tracking” is misleading, and thank them for pointing it out. We have carefully edited the manuscript to use the word “detect” instead to better capture what our method does.

2. The authors only mention how they connect two particles in different frames in the last section of the “Methods”, titled “Measuring particle polarizability”. The expression here is very unclear if the authors want to focus on particle tracking.

We thank the reviewer for pointing this out. We agree that the sudden

inclusion of tracking was not well-explained, especially considering the prior misuse of the word tracking. We have now added the following paragraph to clarify that the linking is done in addition to our method, as well as to motivate why we need to link the particles:

To capture the population distribution of polarizability, we additionally need to link detections of the same particle over time. In this way, each particle is weighed equally in the distribution. We link particles by minimizing a linear sum assignment problem, which was found to be sufficient for this data.

Minor:

1. The normalization factor W' is first mentioned in the "Neural Network Training" section and lacks sufficient explanation.

We thank the reviewer for pointing this out, which is in fact a mistake. The term refers to $\sum D[S(\rho_{m,n}), 0.01]$ in the denominator of the expression for $w_{i,j}$. We have removed W' and instead refer to $\sum D[S(\rho_{m,n}), 0.01]$ directly.

Typos:

1. Page 4, 3rd line: "Theory of ...". Double-quotes.
2. Page 13, 2nd equation of internal consistency loss: the last y_{ij} should be w_{ij} .
3. Page 14, Particle detection criteria: missing period after "we use $\alpha = 0.1$ ".

We thank the reviewer for reading our manuscript carefully enough to notice these typos. In the revised version of the manuscript, we have now remedied these errors in accordance with the reviewer's suggestions.

Reviewer # 2

This paper presents a novel method (LodeSTAR) for particle identification using machine learning that uses unsupervised learning with very small training sets. I find the work to be compelling and the demonstrations of the method to be persuasive. However, I have some concerns about the overall presentation of the manuscript as well as a couple of questions. Therefore I would recommend publication of this article after revisions. My feedback is summarized below:

We thank the reviewer for recognizing the relevance of the work and of the demonstrations. We have now addressed their comments as described below.

1) This may be a requirement of the journal, in which case this comment can be ignored, but I found the placement of the Methods section at the end of the paper to be very frustrating. It should be placed in between the Introduction and Results sections, so that the reader can understand how the procedure works and why before being shown the results.

We sympathize with the frustration of the reviewer. However, as the reviewer conjectured, this format is a requirement of the journal. Nonetheless, to guide the interested reader through the methods section before the main part of the result section, we have added the following introductory paragraph at the start of the results section:

LodeSTAR builds on *geometric deep learning* [13] and the recent surge of self-supervised object tracking methods [14-21] to create a self-supervised (or more precisely, self-distillative) object-detection neural network optimized for microscopy data. Specifically, we exploit the fact that a neural network that is equivariant to rotations and translations (i.e., a neural network for which a roto-translational transformation of the input image produces an equivalent roto-translation of the prediction) operates as an object detector (see Methods, “Theory of geometric self-distillation”). A limitation for general objects is that the exact part of the object that is detected cannot be controlled; however, if the object has a well-defined center (by having at least two axes of symmetry), such a neural network will find the exact center of the object. Building on this insight, we design a novel neural-network architecture that uses global weighted pooling to become inherently translation equivariant (see Methods, “Neural network architecture”). We also design a novel unsupervised training procedure that trains the neural network to become fully roto-translation equivariant. This procedure feeds the neural networks with transformed views of the same image of a single object, and trains it to predict positions that are equivariant with the transformations (see Methods, “Neural network training”).

Moreover, we have rewritten the relevant parts of the methods to make them clearer and more self-contained, so that they can be read without

first having read the results section.

2) The authors present the algorithm as a “tracking” algorithm, but I believe all the demonstrations of the algorithm are for object identification rather than explicit tracking. I.e., objects are identified in single frames, but it was not demonstrated that object A in one frame can be successfully associated with object A in the second frame, and so on for object B, object C, and so forth. Without this demonstration, the method is an identification technique rather than tracking. Object identification is still very valuable, but this distinction is important.

We completely agree with the reviewer that the word “tracking” is misleading. We have carefully edited the manuscript to use the word “detect” instead to better capture what our method does.

3) Related to the previous point, LodeSTAR is based on equivariance of certain properties, primarily invariance under rotation and translation. These are sound for object identification, as demonstrated, but I question their appropriateness for tracking purposes. For tracking rigid particles, these equivariances hold under ideal conditions, but they may not in real situations. The same object visualized twice, especially after motion, may appear differently depending upon the optical diagnostic used. This is true for laser-based scattering, for instance, where a particle’s appearance is highly dependent upon its location and orientation relative to the incident light. For non-rigid objects such as cells, however, does the deformation not become important? Between frames in a sequence, it is possible that a non-rigid object could deform (i.e., change shape) in addition to translation and rotation. What effect would this have on LodeSTAR’s ability to track that object, if indeed the algorithm is capable of tracking between frames rather than just object identification?

We thank the reviewer for this observation. Indeed, LodeSTAR is only capable of object detection, not full frame-to-frame association. We apologize for the confusion our choice of wording has created. As indicated in our answer to the previous question, we now use the word “detection” instead of “tracking” to avoid such confusion.

Reviewer # 3

The authors present a method that belongs to the category of self-supervised techniques (more precisely self-distillation techniques) capable of localizing particles or small simple-shape objects in bioimage analysis without the necessity of creating labels for training. The idea of bringing this type of techniques (that are currently rather intensively researched in the general computer vision community) to bioimage analysis (where supervised techniques still dominate) is surely useful and logical. However, there are a number of serious issues that need to be addressed to make the paper acceptable for publishing.

We thank the reviewer for carefully reviewing our work, and for affording us the opportunity to significantly improve its quality before publication. We agree that self-distillative is a more precise term for our work than self-supervised, but chose the latter since it is more broadly known to the target community. Nonetheless, we feel that the distinction is important, and have added the sentence: *LodeSTAR builds on geometric deep learning[11] and the recent surge of self-supervised object tracking methods [13-20] to create a self-supervised (or more precisely, self-distillative) object-detection neural network [...]*

Major issues:

1) Incorrect terminology including the title and abstract that do not correspond to the paper content at all.

1A) The authors speak about tracking in the whole paper but there is no tracking in the paper, rather there is just detection that precedes tracking. Tracking is about building tracks (or even lineage trees for dividing cells), i.e. inter-connecting detections between frames (along the time axis). The authors use videos (time-lapse image sequences) but perform just detection per frame, no subsequent tracking. They even do not discuss any tracking papers / methods (neither the famous Particle Tracking Challenge – not cited at all). Hence, tracking should be replaced with detection in the whole paper.

We agree with the reviewer that the word “tracking” is misleading and not representative of our work, and sincerely thank them for pointing it out. Following the reviewer’s suggestion, we have carefully edited the manuscript to use the word “detect” instead to better capture what our method does.

1B) The authors mostly speak about particles, sometimes objects. They denote also small objects like cells as particles. Cells are not particles. Particles are nearly point-like objects with few pixel diameters for which segmentation does not make sense. Hence, authors should speak about objects, not particles, starting from the title.

We thank the reviewer for this correction on the definition of the word

“particle”. In fact, we had intended the word particle to refer to a small object. We agree that it is better to explicitly use the wording “small object” to avoid any confusion and we have revised the manuscript accordingly.

1C) The authors should admit right in the title and abstract that they do not speak about all objects but only bioimaging (microscopy) ones that are small (they work with few tens of pixels in diameter) and have rather simple roundish shapes (i.e., not tubular ones, not complex shapes with protrusions, etc.), homogenous interior (no complex texture inside) and, most importantly, are located on a homogenous background (e.g., not cells in tissues). A suitable title might be something like “Single-shot self-supervised small object detection in bioimaging” with a constraint explanation in the abstract.

Following the reviewer’s suggestion, we have now changed the title and abstract to emphasize that LodeSTAR is mainly applicable to small microscopic objects in bioimaging. Furthermore, we have now tested LodeSTAR on more complex and non-homogeneous backgrounds, where LodeSTAR still works, even though its performance is slightly lower than in the case of a homogeneous backgrounds. For example, in Figure R1, we provide an example of the detection of stained nuclei within a relatively complex background represented by a tissue and show that LodeSTAR manages to detect most nuclei.

Figure R1: Detection of cell nuclei in areolar connective tissue.

2) Missing state-of-the-art.

The authors do not cite any relevant papers from the computer vision community on self-supervised (or cell-distillation) object detection (or even segmentation) approaches (with the exception of [22] in one sentence in the conclusion). There are many recent relevant papers, e.g. <https://arxiv.org/abs/2110.14711> (survey with a nice overview Figure)

<https://arxiv.org/abs/2109.14279>

<https://www.biorxiv.org/content/10.1101/2021.01.07.425773v1>

<https://arxiv.org/abs/2010.12023>

We are grateful to the reviewer for providing us with relevant works. We have referenced them in the introduction, along with a few other works that we use later on for comparison. They are primarily referenced in the sentence: LodeSTAR builds on *geometric deep learning*[13] and the recent surge of self-supervised object tracking methods [14-21] to create a self-supervised (or more precisely, self-distillative) object-detection neural network

3) Missing comparison of the developed method with state-of-the-art

The authors compare the developed method (in Fig.1g-k) with just two other rather old methods: trivial centroid approach ([12] from 1996) and radial symmetry centers ([13] from 2012). I do not think this is enough. Some representatives of modern state-of-the-art object detection methods should be added, e.g., self-distillative BYOL-based approaches such as SoCo (see the survey mentioned above). Even if the output of the modern method is a bounding box, it can be converted to the center output by computing the center of the bounding box. It would also be nice to compare the self-supervised approaches to the supervised ones.

In the revised manuscript, we have followed the reviewer’s suggestion and compared the performance of LodeSTAR with more modern methods. We address this concern in two parts.

First, we added a comparison to the theoretical Cramer-Rao lower bound (CRLB), which sets a bound on the localization accuracy possible based on the information in the image. We find that LodeSTAR is able to very closely approach the CRLB, which means that other methods can only outperform LodeSTAR marginally. This information has been added to Figure 1 of the main text (reproduced in Figure R2 in the response). While this bound has been reached by other methods (for example, for point particles by the recently published DECODE method, cited as [9]), this is to our knowledge the first time this feat has been achieved for a wide range of small objects by the same method. We now also mention the CRLB in the main text as:

In fact, we find that LodeSTAR achieves a near optimal performance.

Figure R2: **LodeSTAR single-shot training and performance.** **a** Example image of a single particle used to train the neural network ($N \times M$ pixels, C color channels). **b** Two copies of the original image transformed by translations and rotations. **c** The transformed images are fed to a convolutional neural network. **d** The neural network outputs two tensors (feature maps), each with $N/2 \times M/2$ pixels: One (top) is a vector field where each pixel represents the distance from the pixel itself to the particle (top left, blue arrows), which is then transformed so that each pixel represents the distance of the particle from the center of the image (top right, blue markers). The other tensor (bottom) is a weight map (normalized to sum to one) corresponding to the contribution of each element in the top feature map to the final prediction. **e** These two tensors are multiplied together and summed to obtain a single prediction of the position of the particle for each transformed image. **f** The predicted positions are then converted back to the original image by applying the inverse translations and rotations. The neural network is trained to minimize the distance between these predictions. **g-k** LodeSTAR performance on $64 \text{ px} \times 64 \text{ px}$ images containing different simulated particle shapes: **g** point particle, **h** sphere, **i** annulus, **j** ellipse, and **k** crescent. Even though LodeSTAR is trained on a single image for each case (found in the corresponding inset), its root mean square error (RMSE, blue circles) **approaches the Cramer-Rao lower bound, and significantly outperforms** traditional methods based on the centroid [22] (gray triangles) or radial symmetry [23] (gray diamonds), especially at low signal-to-noise ratios (SNRs). Interestingly, even in the crescent case **k**, where there is no well-defined particle center, LodeSTAR is able **locate** it to within a tenth of pixel.

We evaluate the optimal performance by calculating the Cramer-Rao (CR) lower bound on the localization error [24] (see Methods “Evaluating the Cramer-Rao lower bound” for more details). The CR lower bound defines the optimal performance any estimator can achieve based on the information content in the image. LodeSTAR manages to match the CR lower bound for most SNRs, only falling short for very low SNRs (Figure 1g)

We also added a “Methods” section further detailing how the CRLB was calculated, mainly based on theory from [24]:

For a pure Poisson process, the Fisher information matrix is calculated as

$$\mathbf{I}(\theta) = \sum_{k=1}^K \left[\left(\frac{\partial \nu_{\theta,k}}{\partial \theta} \right)^T \left(\frac{\partial \nu_{\theta,k}}{\partial \theta} \right) \frac{1}{\nu_{\theta,k}} \right],$$

where k enumerates K pixels in the image, θ is the parameter vector, and $\nu_{\theta,k}$ is the expected number of photons for a given pixel and detector [24]. Since the images are all synthetic, $\nu_{\theta,k}$ is known exactly.

We considered the x and y position of the particle, as well as the orientation α as the parameters. To calculate the partial derivatives, we simulated $\nu_{\theta,k}$ at 10 times higher resolution than the test images and further interpolated the spacing between the pixels. Finally, the bounds on the variance of an estimator are calculated as the diagonal elements of the inverted information matrix, i.e.: $[\delta_x^2, \delta_y^2, \delta_\alpha^2] = \mathbf{I}^{-1}(\theta)$. The bound on the root mean squared error is finally calculated as $\sqrt{\frac{\delta_x^2 + \delta_y^2}{2}}$.

Second, we compare LodeSTAR with several deep learning methods for object detection. We mainly focus on supervised methods, since modern self-supervised methods for object detection are not designed to extract sub-pixel positions and, therefore, the comparison would be unfair as the subpixel accuracy of LodeSTAR would easily outperform them. (We will discuss the comparison to modern self-supervised/self-distillative object detection methods in the next point, focusing on detection quality.) Moreover, we only consider methods that can learn to detect any object, instead of specializing to small subsets, such as single-molecule emitters or Mie-spheres. We chose four supervised object detection methods and one self-supervised method that exemplify a few approaches for locating objects:

1. **SoCo** for a mostly unsupervised bounding-box approach (ref. [26] in the main manuscript).
2. **YOLOv4** for fully supervised bounding-box approach (ref. [25] in the main manuscript).
3. **DeepTrack 1.0** for locating single particles (ref. [8] in the main manuscript).

Figure R3: **Evaluation positioning accuracy of object detection methods.** The root mean squared error of the position accuracy for six methods, a CNN with a dense top (described in [8]), YOLOv4[25], SoCo[26], a segmentation CNN (described in [7]), LodeSTAR* (which is the architecture of LodeSTAR trained in a supervised manner), and LodeSTAR. We evaluate these over several a range of sizes of training sets, from 1 datapoint to 1000 datapoints, on five shapes: **a** a point particle, **b** a spherical particle, **c** an annulus, **d** an ellipse, and **e** a crescent moon shape. LodeSTAR outperforms all other methods at all sizes of training sets. In fact, LodeSTAR reaches optimal performance using just one datapoint for training.

4. **Segmentation-based convolutional neural network** for another translation-equivariant approach (ref. [7] in the main manuscript).
5. **Supervised LodeSTAR**, which is the LodeSTAR neural network but trained using a normal supervised approach.

Since the methods are not designed to be few-shot, we evaluated the models on increasing training dataset sizes. We find that regardless of dataset size, LodeSTAR performs better than the first four methods, and equally well as the supervised version of LodeSTAR. See Figure S1 (reproduced here has Figure R3).

As nearly all the real (non-synthetic) datasets used in the paper are taken from the Cell Tracking Challenge [11] (Fig.2 f-h), it would be useful to compare the detection results with those listed in the current leaderboard for detection (DET measure of Cell Segmentation Benchmark). To measure DET, it would be necessary to replace center results with some circular markers (of reasonable size corresponding to average object size) to ensure overlap with at least half of the markers in [11] but should be doable. Currently, the performances on real datasets in Fig.2 are not compared to any other methods at all!

Following the reviewer’s suggestion, we computed the DET score as well as the F1 score in all cases. Although the comparison to the public dataset is still not completely valid since the test set is not available (reasonably so), we find that our performance is comparable with the top results in the challenge. This has been added to the main text:

We also compare LodeSTAR to published results in the cell-tracking

challenge [12]. We measure the DET* metric, where the * indicates that it is a version of the DET score [31] that supports object detection methods that do not segment the objects (see Methods, “Object detection comparison”). We find DET*-scores of 0.989, 0.952 and 0.936 respectively; all of which are comparable to the top scores on the official benchmark [32].

LodeSTAR achieves these results despite being trained on more than 1000 times less data than the published methods. Although the comparison to the published methods cannot be made exactly since the official test set is not published (and the possibility of discrepancies between *DET* and *DET**), these results demonstrate that LodeSTAR is at least comparable to state-of-the-art supervised object detection methods.

We also evaluated four self-supervised methods (SoCo, FSDet, DETReg and InstanceLoc) on the same datasets. They were chosen primarily based on how easily they could be made to train on custom datasets. Since these methods all rely on extensive pretraining, we tried both using their published pretrained method and pretraining from scratch on the cell tracking datasets. We found that using the provided pretrained models was the superior of these two approaches in all cases. They were fine-tuned on a small subset of the dataset containing two representative cell-instances. To avoid concerns about overtraining, we monitored the F1-score on the test set after every epoch and chose the optimal cutoff-point. Despite these advantages, LodeSTAR proved significantly superior in terms of F1-score for the chosen datasets, as summarized by Table R1 (Table 2 in revised main text), accompanied with the following new paragraphs:

To compare the performance of LodeSTAR with other self-supervised methods, we construct an evaluation method. This method first compares predicted positions with publicly available annotations for the first three datasets [12]. It marks cells as found if a predicted position overlaps with the segmentation of the cell. Using this method, we evaluate the F1-score of LodeSTAR, as well as those of four other self-supervised object detection methods: SoCo [26], FSDet [27], InstanceLoc [28], and DETReg [29]. See Methods, “Object detection comparison” for more details.

F1-score	BF-C2DL-HSC	Fluo-C2DL-Huh7	PhC-C2DL-PSC
SoCo	0.85	0.84	0.48
FSDet	0.73	0.75	0.59
InstanceLoc	0.82	0.47	0.63
DETReg	0.71	0.61	0.44
LodeSTAR	0.98	0.97	0.97

Table R1: F1-score of self-supervised and low-shot object detection methods evaluated on the datasets in the cell tracking challenge [12]. LodeSTAR significantly outperforms the other methods in each experiment.

Table 2 summarizes the results of the comparison. We find that LodeSTAR achieves results far superior to the alternative methods. We can also pinpoint the reasons for failure of the alternative methods. For the dataset in Figure 2f, we find that they all perform well when the number of cells is low, but fail when the number of cells increase above some critical threshold. For the dataset in Figure 2g, we find that they struggle at finding both the small and the large cells simultaneously given the small training sets. For the dataset in Figure 2h, we find that they struggle to generalize at all beyond the training set, most likely because of the variability of the cells' morphology.

4) Problems with metrics.

The authors use RMSE for single simple synthetic object studies in Fig.1, which is fine, but then use TPR+FDR for real datasets, which is not compatible with other studies and causes the problem of not comparing the developed method performance to any other method (as mentioned above). Most other detection studies typically use F1-Score (or AP for probabilistic outputs of neural networks). If subsequent tracking is anticipated (which is the case of videos here), then recall is favored to precision like in [11] where DET measure (similar to F3-Score) is used. Anyway, the authors should use a measure compatible with other studies (or should measure TPR+FDR for other methods, which is more time-consuming).

We agree with the reviewer's suggestion to use metrics that facilitate comparisons to other methods. Therefore, in the revised manuscript, we have evaluated the F1-score and DET-score for Fig 2, and used these metrics in place of TPR and FPR.

Also for holography experiments (Figure 4e) some metric should be used – here the detection performance (for blue and orange markers) is not measured here at all.

Since these are experimental datasets, we do not have absolute ground truths. However, holography can reasonably be simulated with Mie theory. As such we have synthetically replicated our setups (both for Figure 3 and Figure 4) and evaluated F1-score as a function of signal-to-noise ratio. We find that for all holographic experiments in the paper, the F1 score is above 0.99. This has been added to the supplementary material as Figure S2, here reproduced as Figure R4. The following sentences was also added to the main text:

The particles were both detected and located in 3D space using LodeSTAR. Using a synthetic dataset replicating the experimental conditions, we find that LodeSTAR achieves an F1-score of 0.99 (see Supplementary Figure S2a for details). [...] First, we measure LodeSTAR's ability to detect particles using a synthetic replica of the optical setup. We find

Figure R4: **Detection accuracy of LodeSTAR on holographic data.** The detection F1 score as a function of the signal-to-noise ratio (calculated as $\max(|I|)/\sqrt{\sigma_{Re}^2 + \sigma_{Im}^2}$, where I is the field, and σ_{Re}, σ_{Im} are the standard deviations of the real and imaginary parts of the field), on synthetic data. **a** F1-score (shaded region is the 95-th quantile) of LodeSTAR on the standard holographic setup. For the two particle sizes used, the F1 score is around 0.98. **b** F1-score (shaded region is the 95-th quantile) of LodeSTAR on the sample-position-modulated holographic setup. For the particle size used, the F1 score is around 0.96.

that LodeSTAR achieves an expected F1-score of over 0.95 (see Supplementary Figure S2b).

5) Unclear TP definition.

The authors have not even described how they define TP (and consequently FP and FN) for the computation of TPR+FDR. Usually, some threshold distance is chosen to match the predicted center position with the ground truth center position but even in this case, it is necessary to solve ambiguities (like multiple predictions matching one ground truth object or vice versa). Also, in [11] markers are used instead of single positions, did the authors convert markers to positions?

We thank the reviewer for allowing us to better explain our methodology. We define TP based on whether the particle overlaps with the segmentation of the cell. We have now described this in Methods as follows:

If a single prediction overlaps with the segmentation of a cell, that prediction is considered a true positive. If multiple predictions overlap, each additional prediction is considered a false positive. Each prediction not overlapping with a segmentation is considered a false positive, and each

segmentation with no corresponding prediction counts as a false negative.

6) 3D capabilities unclear.

It is not clear whether the method is applicable only in 2D or also in 3D. From the abstract, it seems 3D is addressed as well: “We also exploit additional symmetries to extend the measurable particle properties to the particle’s vertical position”. However, the authors use 2D examples (and descriptions) throughout the paper with the exception of the 3D holography example. What about classical (non-interference) 3D images?

We thank the reviewer for pointing out this possible source of misinterpretation. Indeed, we focus on utilizing Fourier transformations to extract 3D positions from 2D holographic images. This is only possible for special optical setups, such as holography.

Extracting 3D positions from 3D images, such as from light-sheet microscopy, is possible under the same framework as this work (by extending the transformations to 3D rotations and translations). However, it is not something we cover in this manuscript. We have modified the text to make this clearer by explicitly mentioning that this works for interference microscopy: [...] the particle vertical position in interference holography exploiting the propagation symmetry of the image in Fourier space [...]

7) Unclear roto-translational symmetries.

Already in the abstract, the authors speak about “exploiting the inherent roto-translational symmetries of the data” as the basis of their method. Also, the very first section of Methods is devoted to symmetries as the basis of their method, here they define assumptions (1)-(4). However, objects in the experimental data often do not exhibit such symmetries, not even those objects used in the paper in Figure 2gh – they are not rotationally symmetric. For real world the assumptions (1)-(4) simply do not hold: (1) because h is not ideal (there is some error remaining after optimization), (2) because one cell is not just equal to rotated and translated another cell, (3)-(4) is admitted by the authors as not fulfilled for asymmetrical objects. So, there is an inconsistency between the theory behind the method and the properties of experimental objects. This inconsistency should be addressed somehow. If the method really works for “arbitrary shapes” (as mentioned on p.5), then it should be explained why.

We thank the reviewer for this comment that made clear that what we refer to as symmetries might be confusing. We believe that this confusion stems from us not clearly distinguishing symmetries in the data and symmetries in the network. Summarizing, a network with roto-translation equivariance for some reference object, *regardless of the symmetries of the object itself*, is sufficient for the network to detect identical objects. Likely due to the smoothness of the neural network, this also extends to

morphologically similar objects. If the detected object has also symmetries, this serves to further constrain the final model to predict exactly the centroid of the object. We have added the following sentences to make this more clear: Specifically, we exploit the fact that a neural network that is equivariant to rotations and translations (i.e., a neural network for which a roto-translational transformation of the input image produces an equivalent roto-translation of the prediction) operates as an object detector (see Methods, “Theory of geometric self-distillation”). A limitation for general objects is that the exact part of the object that is detected cannot be controlled; however, if the object has a well-defined center (by having at least two axes of symmetry), such a neural network will find the exact center of the object.

Moreover, as the reviewer notes, the performance of the method cannot be explained by this theory alone. Overall, our intent with the theory is to show one specific ideal case where we can prove that the method works, and then demonstrate that the method works for non-ideal cases by means of empirical evidence. In other words, our theory is not of the “if-and-only-if” kind. We now completely revised the theory section to make it clearer when it holds, what parts are necessary for LodeSTAR, and what parts only contribute to constraining LodeSTAR further. Specifically, we have explicitly constrained the theory to the ideal case, and separated it into three parts. First we show that a perfectly equivariant network finds an identical object consistently. Second, we show that if the object also has symmetries, we can better constrain what part of the object LodeSTAR will find. Third, we demonstrate that small errors in the equivariance of the neural network results in small errors in the performance. We also made the theory easier to read by focusing on how it relates to finding the position of an object, while adding a paragraph at the end of the text that shows how to generalize it to other properties. This is the revised section:

This section shows theoretically that LodeSTAR locates objects correctly. First, we consider the case of an arbitrary, constant object with no noise. Then, we show the special case of a symmetrical object. Finally, we consider an imperfect estimator. We will describe the theory to find the position of the particle, but the same arguments easily extend to other properties for which a symmetry is available (e.g., mass, size, orientation).

Let X be the set of possible images of an object, Y be the set of possible positions of the object in the image, f be the ground-truth function that returns the particle position and h be a neural network, both mapping $X \rightarrow Y$, and G be the Euclidean group consisting of translations, rotations and reflections, acting on both X and Y .

Given a single image $x_0 \in X$, we can define the subset X^{x_0} , which are all the elements of X reachable by acting on x_0 with G . f is, by definition, equivariant to G on the subset X^{x_0} .

Now, additionally assume that (Assumption 1), h is trained to be

equivariant to G on the subset X^{x_0} , i.e., $h(gx) = gh(x) \forall g \in G$ and $\forall x \in X^{x_0}$. Next, we defined the error \mathbf{c} as the error $f(x_0) - h(x_0)$ for the input image x_0 . Now, for any other $x' \in X^{x_0}$, we know that $x' = g'x_0$ for some $g' \in G$. Let us factor g' into $g'_r g'_t$, where g'_t is the translational part of the transformation, and g'_r is the non-translational remainder. Finally, we define h as “consistent” if $f(x') - h(x') = g'_r \mathbf{c}$ (i.e., the offset is independent of the translation and transforms correctly with g'_r). We can easily show this to hold by substitution:

$$f(x') - h(x') = f(g'x_0) - h(g'x_0) \quad (1)$$

$$= g'_r g'_t f(x_0) - g'_r g'_t h(x_0) \quad (2)$$

$$= g'_r f(x_0) - g'_r h(x_0) \quad (3)$$

$$= g'_r [f(x_0) - h(x_0)] \quad (4)$$

$$= g'_r [f(x_0) - f(x_0) + \mathbf{c}] \quad (5)$$

$$= g'_r \mathbf{c} \quad (6)$$

Note that g'_t disappears in Eq. (3) since it equals to adding and subtracting a constant vector. In Eq. (4), we use the fact that g'_r is linear.

This is sufficient to show that LodeSTAR, trained to be perfectly roto-translationally equivariant to some input image x_0 , learns to detect identical objects consistently. However, for two different subsets X^{x_0} and X^{x_1} , \mathbf{c} may have different values. Consequently, internal consistency is not guaranteed between these two sets. If an action g^* can be defined that can transform some element of either set to the other, then for the combined set $G \cup g^*$, $X^{x_0} \equiv X^{x_1}$ by definition. LodeSTAR is then guaranteed to be internally consistent for both subsets.

Now we demonstrate what happens if the images have some symmetry with respect to the group G . To do this, we assume that (Assumption 2) there exists a transformation $g \in G$ that is not a trivial transformation, such that $gx = x$ for some $x \in X^{x_0}$. This is known as a symmetry.

We know that $f(x') - h(x') = g'_r \mathbf{c}$. Now, due to Assumption 2, we know that there exists some non-trivial g^* such that $g^*x = x$. Consequently,

$$g'_r \mathbf{c} = f(x') - h(x') \quad (7)$$

$$= f(g^*x') - h(g^*x') \quad (8)$$

$$= g^* f(x') - g^* h(x') \quad (9)$$

$$= g_r^* g_t^* f(x') - g_r^* g_t^* h(x') \quad (10)$$

$$= g_r^* f(x') - g_r^* h(x') \quad (11)$$

$$= g_r^* [f(x') - h(x')] \quad (12)$$

$$= g_r^* g_r' \mathbf{c} \quad (13)$$

In other words, we have constrained \mathbf{c} such that $g'_r \mathbf{c} = g_r^* g_r' \mathbf{c}$ holds. How this constraint manifests depends on the transformation g_r^* . If g_r^*

is equivalent to a reflection along some line, then $g'_r \mathbf{c}$ must be parallel to that line. This is equivalent to saying that the offset between f and h in the direction normal to a reflection symmetry must be 0. If g_r^* is some rotation about a point, then \mathbf{c} must be zero, since $y = g_{\text{rotation}} y$ can only hold if g_{rotation} is a trivial rotational transformation (i.e., rotating an integer multiple of 2π), or if $y = \mathbf{0}$.

We find that any rotational symmetry necessarily means that $f = h$ on X^{x_0} . In fact, we have done so without defining f more exactly than being equivariant to G . Further, we can show that f and h necessarily find the center of rotational symmetry. Consider an x that is symmetric to a purely rotational transformation, $g_{\text{rotation}} x = x$. Then $h(x) = h(g_{\text{rotation}} x) = g_{\text{rotation}} h(x)$. This can only be true if $h(x)$ is the center of rotation, for the same reason that \mathbf{c} had to be 0.

In reality, Assumption 1 will never hold exactly. LodeSTAR will not be perfectly equivariant to the Euclidean group. As such, it is important to verify that a small deviation from equivariance results in a small error in predicted position. We assume that $h(gx) = gh(x) + \mathbf{e}_{x,g}$ for some small $\mathbf{e}_{x,g}$. We recalculate Eq.(1)-(6) with this new assumption.

$$f(x') - h(x') = f(g'x_0) - h(g'x_0) \quad (14)$$

$$= g'_r g'_t f(x_0) - g'_r g'_t h(x_0) - \mathbf{e}_{x',g'} \quad (15)$$

$$= g'_r f(x_0) - g'_r h(x_0) - \mathbf{e}_{x',g'} \quad (16)$$

$$= g'_r [f(x_0) - h(x_0)] - \mathbf{e}_{x',g'} \quad (17)$$

$$= g'_r [f(x_0) - f(x_0) + \mathbf{c}] - \mathbf{e}_{x',g'} \quad (18)$$

$$= g'_r \mathbf{c} - \mathbf{e}_{x',g'} \quad (19)$$

Then, substituting this into Eq. (7)-(13), we find that

$$g'_r \mathbf{c} = f(x') - h(x') + \mathbf{e}_{x',g'} \quad (20)$$

$$= f(g^* x') - h(g^* x') + \mathbf{e}_{x',g'} \quad (21)$$

$$= g^* f(x') - g^* h(x') + \mathbf{e}_{x',g'} \quad (22)$$

$$= g_r^* g_t^* f(x') - g_r^* g_t^* h(x') + \mathbf{e}_{x',g'} \quad (23)$$

$$= g_r^* f(x') - g_r^* h(x') + \mathbf{e}_{x',g'} \quad (24)$$

$$= g_r^* [f(x') - h(x')] + \mathbf{e}_{x',g'} \quad (25)$$

$$= g_r^* [f(x') - h(x') + \mathbf{e}_{x',g'} - \mathbf{e}_{x',g'}] + \mathbf{e}_{x',g'} \quad (26)$$

$$= g_r^* g_r' \mathbf{c} + \mathbf{e}_{x',g'} - g_r^* \mathbf{e}_{x',g'} \quad (27)$$

Note that in Eq. (22), we can omit the \mathbf{e}_{x',g^*} since $x = g^* x$, so \mathbf{e}_{x',g^*} must be 0. We see that $g'_r \mathbf{c} - g_r^* g_r' \mathbf{c} = \mathbf{e}_{x',g'} - g_r^* \mathbf{e}_{x',g'}$. Remembering that g_r^* acts on R^2 like a matrix, we see that $(\mathbf{I} - g_r^*) g_r' \mathbf{c} = (\mathbf{I} - g_r^*) \mathbf{e}_{x',g'}$. If $(\mathbf{I} - g_r^*)$ is invertible (as is the case for rotations), we find that $g'_r \mathbf{c} = \mathbf{e}_{x',g'}$. In other words, the difference between f and h is equal to the error associated with imperfect equivariance. If this error is small, then $g'_r \mathbf{c}$ is also small.

If $(\mathbf{I} - g_r^*)$ is not invertible (as is the case for reflections), we need to consider its eigenvalues. In particular (for convenience), its left eigenvalues. For any left eigenvector \mathbf{v}_λ with a eigenvalue λ , we find that

$$(\mathbf{I} - g_r^*)g'_r\mathbf{c} = (\mathbf{I} - g_r^*)\mathbf{e}_{x',g'} \quad (28)$$

$$\mathbf{v}_\lambda^T(\mathbf{I} - g_r^*)g'_r\mathbf{c} = \mathbf{v}_\lambda^T(\mathbf{I} - g_r^*)\mathbf{e}_{x',g'} \quad (29)$$

$$\lambda\mathbf{v}_\lambda^Tg'_r\mathbf{c} = \lambda\mathbf{v}_\lambda^T\mathbf{e}_{x',g'} \quad (30)$$

If $\lambda = 0$, this reduces to $0 = 0$, which does not restrict $g'_r\mathbf{c}$. If $\lambda \neq 0$, then we see that the projection of \mathbf{c} onto \mathbf{v}_λ is equal to the projection of $\mathbf{e}_{x',g'}$ onto \mathbf{v}_λ . The error in this direction must then be small if $\mathbf{e}_{x',g'}$ is small.

Finally, it should be noted that to generalize this argument to other properties one needs to be careful of trivial solutions. For example, in the absence of a translation-like transformation, there will often be a trivial solution at $h \equiv \mathbf{0}$. Symmetries to G can only exist if the center of symmetry is at origin, whereby $h \equiv \mathbf{0}$ is a correct, yet woefully uninformative solution. As a rule of thumb, a group consisting of one translation-like transformation is sufficient for LodeSTAR to be internally consistent, while a group that additionally has some linear transformation (with a corresponding symmetry) is sufficient to ensure that there is only one optimal solution.

8) The mathematical symbols and equations are not properly defined. The mathematical symbols and equations are not properly defined. In many places in Methods, there are symbols in equations or text that are not defined or used interchangeably with sub-indices and without them, which makes it hard to understand the theory. Moreover, there are no accompanying figures visualizing the description (just a very poor network symbol in Figure 1c with no details).

We have rewritten the sections in question to improve readability and consistency of notation. Below we address the changes to each section individually.

Neural network architecture: undefined x , y , ρ , i , j , unclear what precisely is at the input, which weights are referred to (there are many weights in each network), why N is size without tensors and M with tensors, etc.

We have now clearly defined Δx , Δy and ρ as two-dimensional tensors, which are also the channels of the output of the neural network. We have added the missing indices in the equations for x and y . We have now correctly defined both M and N as referring to the input. We have removed references to ρ and w as weights to avoid confusion with internal

weights of LodeSTAR. We have also added some text motivating the calculations and demonstrating how they result in translation equivariance.

Neural network training: sums without bounds (not clear from which value to which the sum runs), undefined $W_{i,j}$ (used after sum), x^{ω} with missing sub-indices and missing bounds for summation, etc.

We have restructured this section completely to allow it to be read more independently from the main text. We have better defined the sums, and introduced a more consistent notation.

Particle detection criteria: undefined K (Number of clusters? How is it determined?), the sum over k should probably be applied to the whole remaining part, not just first parentheses (i.e., additional parentheses required), $\alpha = 1$ for three cases means $\beta = 0$, i.e. clustering not applied at all – deserves a comment.

We have made the notation consistent with previous sections, and in the text defined K (now C), as the number of channels in the output of the neural network. We have added “A high α is used for the last few examples since the cells are not very similar, which typically leads to poorer clustering performance.” to explain the comparably high α in the three experimental studies. From Table 1 in the main text it can be seen that LodeSTAR clusters predictions much better when objects are similar to the training sample. Consequently, we chose a high α under the assumption that the clustering metric would not be very useful.

Measuring particle polarizability: all symbols undefined and no citation. At least Clausius-Mossotti relation deserves a citation.

We have properly defined the symbols and added a citations for the Clausius-Mosotti relation

Minor issues: There are some typos in the paper, e.g., “centriod” instead of “centroid” twice, “results of achieved with” (p.12), “f and h be functions h” (p.12), “in in” (p.12), “are and” (p.14).

We have fixed the mentioned typos along with a few others.

REVIEWERS' COMMENTS

Reviewer #1 (Remarks to the Author):

I think most of my concerns are addressed.
The quality of the paper can meet the bar now.

Reviewer #2 (Remarks to the Author):

I appreciate the authors' efforts to address my comments. I feel they have done an admirable job of doing so, and the manuscript is now ready for publication.

Reviewer #3 (Remarks to the Author):

The authors have thoroughly revised the paper; I am satisfied with the way they addressed my issues. Especially, I appreciate additional experiments to compare their method to other methods as well as to the CR lower bound. They also changed the terminology and rewrote unclear parts. I think the paper is understandable now and worth publishing. I have just found a few minor issues that do not require an additional check by reviewers:

- 1) Figure 1d legend: in a vector field, each vector contains not only distance but also direction. Hence, the word “distance” should be replaced with “direction and distance”. The same for the main text. Also, there is a discrepancy between the legend and the main text: in the legend, the position is computed relative to the image center while in the main text it is computed relative to the upper left corner of the image. Please unify.
- 2) Particle detection criteria section: “object vary” instead of “objects vary”.
- 3) Object detection comparison section: typo “where is” instead of “where D is”; also the definition of DET has wrong typesetting – in the PDF, there is just a right bracket in the denominator instead of D_0 and “1-“ is missing before the minimum operation.

Response to the Reviewers' reports on the manuscript entitled **“Single-shot self-supervised object detection in microscopy”** by Benjamin Midtvedt, Jesús Pineda, Fredrik Skärberg, Erik Olsén, Harshith Bachimanchi, Emelie Wesén, Elin K. Esbjörner, Erik Selander, Fredrik Höök, Daniel Midtvedt, and Giovanni Volpe

Reviewer # 1

I think most of my concerns are addressed. The quality of the paper can meet the bar now.

We thank the reviewer for the positive reception of our revision.

Reviewer # 2

I appreciate the authors' efforts to address my comments. I feel they have done an admirable job of doing so, and the manuscript is now ready for publication.

We thank the reviewer for the positive reception of our revision.

Reviewer # 3

The authors have thoroughly revised the paper; I am satisfied with the way they addressed my issues. Especially, I appreciate additional experiments to compare their method to other methods as well as to the CR lower bound. They also changed the terminology and rewrote unclear parts. I think the paper is understandable now and worth publishing. I have just found a few minor issues that do not require an additional check by reviewers:

We thank the reviewer for the positive reception of our revision as well as for the additional comments below.

1) Figure 1d legend: in a vector field, each vector contains not only distance but also direction. Hence, the word “distance” should be replaced with “direction and distance”. The same for the main text. Also, there is a discrepancy between the legend and the main text: in the legend, the position is computed relative to the image center while in the main text it is computed relative to the upper left corner of the image. Please unify.

We have made these corrections.

2) Particle detection criteria section: “object vary” instead of “objects vary”.

We have made this correction.

3) Object detection comparison section: typo “where is” instead of “where D is”; also the definition of DET has wrong typesetting – in the PDF, there is just a right bracket in the denominator instead of D_0 and “1-“ is missing before the minimum operation.

We have made this correction.